# The RNA helicase DDX39B activates FOXP3 RNA splicing to control T regulatory cell fate

**Minato Hirano[1,2†], Gaddiel Galarza-Muñoz[1,3†], Chloe Nagasawa[1,4], Geraldine Schott[1‡], Liuyang Wang[5], Alejandro L Antonia[5], Vaibhav Jain[6], Xiaoying Yu[1,7], Steven G Widen[1], Farren BS Briggs[8], Simon G Gregory[5,6,9], Dennis C Ko[5,10], William S Fagg[1,11], Shelton Bradrick[12*§], Mariano A Garcia-Blanco[1,13,14*]**

[1]Department of Biochemistry and Molecular Biology, University of Texas Medical Branch, Galveston, United States; [2]National Research Center for the Control and Prevention of Infectious Disease, Nagasaki University, Nagasaki, Japan; [3]Autoimmunity Biological Solutions, Galveston, United States; [4]Human Pathophysiology and Translational Medicine Program, Institute for Translational Sciences, University of Texas Medical Branch, Galveston, United States; [5]Department of Molecular Genetics and Microbiology, Duke University, Durham, United States; [6]Duke Molecular Physiology Institute, Duke University, Durham, United States; [7]Department of Preventive Medicine and Population Health, University of Texas Medical Branch, Galveston, United States; [8]Department of Population and Quantitative Health Sciences, School of Medicine, Case Western Reserve University, Cleveland, United States; [9]Department of Neurology, Duke University School of Medicine, Durham, United States; [10]Division of Infectious Diseases, Department of Medicine, Duke University, Durham, United States; [11]Transplant Division, Department of Surgery, University of Texas Medical Branch, Galveston, United States; [12]Institute of Human Infections and Immunity, University of Texas Medical Branch, Galveston, United States; [13]Department of Internal Medicine, University of Texas Medical Branch, Galveston, United States; [14]Department of Microbiology, Immunology and Cancer Biology, University of Virginia, Charlottesville, United States

*For correspondence:
ssbradri@utmb.edu (SB);
maragarc@utmb.edu (MAG-B)

[†]These authors contributed equally to this work

**Present address:** [‡]Quivira Trce, League, United States; [§]The Trudeau Institute, Algonquin Ave, New York, United States

**Abstract** Genes associated with increased susceptibility to multiple sclerosis (MS) have been identified, but their functions are incompletely understood. One of these genes codes for the RNA helicase DExD/H-Box Polypeptide 39B (DDX39B), which shows genetic and functional epistasis with interleukin-7 receptor-α gene (*IL7R*) in MS-risk. Based on evolutionary and functional arguments, we postulated that DDX39B enhances immune tolerance thereby decreasing MS risk. Consistent with such a role we show that DDX39B controls the expression of many MS susceptibility genes and important immune-related genes. Among these we identified *Forkhead Box P3* (*FOXP3*), which codes for the master transcriptional factor in CD4⁺/CD25⁺ T regulatory cells. DDX39B knockdown led to loss of immune-regulatory and gain of immune-effector expression signatures. Splicing of *FOXP3* introns, which belong to a previously unrecognized type of introns with C-rich polypyrimidine tracts, was exquisitely sensitive to DDX39B levels. Given the importance of FOXP3 in autoimmunity, this work cements DDX39B as an important guardian of immune tolerance.

## Editor's evaluation

DDX39B is a helicase with known functions in mRNA splicing and nuclear export. This important study provides convincing evidence that DDBX39B regulates Foxp3, a lineage marker for T-regulatory cells in the immune system. The work provides a detailed analysis of the post-transcriptional regulation of Foxp3, and also positions DDX39B more broadly, as an important player in the regulation of autoimmune responses. The work will be of interest to RNA biologists, immunologists, and those studying autoimmune disorders.

## Introduction

Autoimmune diseases are caused by a combination of environmental and genetic factors. Genetic factors associated with increased susceptibility to multiple sclerosis (MS), an autoimmune disease of the central nervous system, have been identified, but their mechanisms of action are incompletely understood (*Briggs, 2019*; *Patsopoulos et al., 2019*). We established that the association between MS risk and the interleukin-7 receptor-α gene (*IL7R*) is mediated by alternative splicing of *IL7R* transcripts (*Gregory et al., 2007*). The disease associated allele of the single-nucleotide polymorphism (SNP) rs6897932 in exon 6 of IL7R strengthens an exonic splicing silencer (ESS) and increases skipping of exon 6 leading to increased production of soluble IL7R (sIL7R) (*Evsyukova et al., 2013*; *Gregory et al., 2007*; *Schott et al., 2021*). Elevated levels of sIL7R have been shown to exacerbate the inducible murine MS-like model experimental autoimmune encephalomyelitis (EAE) and are proposed to increase the bioavailability of IL7 (*Lundström et al., 2013*). Given the importance of exon 6 splicing we identified RNA binding proteins that bind and impact its splicing, and among these was the RNA helicase DExD/H-Box Polypeptide 39B (DDX39B; *Evsyukova et al., 2013*; *Galarza-Muñoz et al., 2017*).

DDX39B, known to RNA biologists as U2AF65-Associated Protein 56 KDa (UAP56), plays roles in RNA splicing and nucleocytoplasmic transport (*Shen, 2009*), but was first discovered by immunologists, who named it HLA-B Associated Transcript 1 (BAT1) (*Spies et al., 1989*) and linked it to autoimmune diseases (*Degli-Esposti et al., 1992*). We connected the RNA biology and immunology roles of DDX39B showing that it activates splicing of IL7R exon 6 and represses production of sIL7R (*Galarza-Muñoz et al., 2017*). Furthermore, *DDX39B* shows genetic and functional epistasis with *IL7R* in enhancing MS risk (*Galarza-Muñoz et al., 2017*). Based on these studies, we proposed that DDX39B plays protective roles in MS and other autoimmune diseases (*Galarza-Muñoz et al., 2017*). The alternative splicing of IL7R exon 6 is conserved only in primates, while *DDX39B* is conserved and located in the major histocompatibility complex (MHC) class III region in all vertebrates (*Schott and Garcia-Blanco, 2021*), suggesting that DDX39B controls important immune modulators other than IL7R.

Here, we show that DDX39B controls expression of gene products involved in autoimmunity including Forkhead Box P3 (FOXP3), a master regulator of the development, maintenance and function of CD4$^+$/CD25$^+$ regulatory Tcells (Tregs; *Georgiev et al., 2019*; *Hori, 2021*; *Josefowicz et al., 2012*), and a repressor of autoimmune diseases (*Bennett et al., 2001*; *Brunkow et al., 2001*; *Chatila et al., 2000*; *Wildin et al., 2001*). Splicing of FOXP3 introns, which have C-rich polypyrimidine tracts, is strongly dependent on DDX39B, making FOXP3 expression highly sensitive to the levels of this RNA helicase.

## Results

### DDX39B controls expression of genes associated with multiple sclerosis risk

We had previously knocked down DDX39B with two independent shRNAs (Sh3 and Sh5) in primary human CD4$^+$ T cells from six healthy human donors (Donors 1–6) and shown that DDX39B depletion led to increased skipping of IL7R exon 6 (*Galarza-Muñoz et al., 2017*). To identify other RNAs affected by DDX39B in these immune relevant cells, we carried out RNA sequencing (RNAseq) of polyadenylated RNA from control and DDX39B depleted CD4$^+$ T cells from two of these donors (donors 1 and 4). More transcript level changes were detected with Sh3 treatment than with Sh5

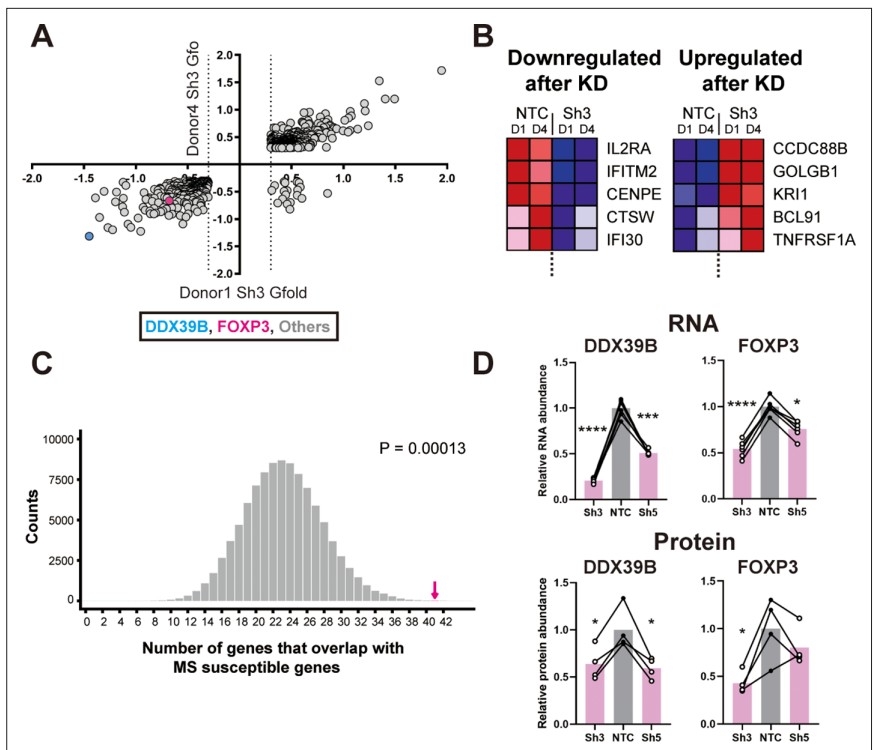

**Figure 1.** Loss of FOXP3 expression in DDX39B-depleted T cells. (**A**) RNA abundance changes between control (NTC) and DDX39B-depleted (Sh3) CD4+ T cells from two healthy individuals (Donor 1 and Donor 4) identified by RNAseq. Data points for DDX39B (cyan) and FOXP3 (magenta) are indicated. (**B**) Examples of MS susceptibility genes differentially expressed upon DDX39B knockdown. The heat map shows expression of five genes between control (NTC) and DDX39B depleted (Sh3) CD4+ T cells. (**C**) Enrichment analysis of MS susceptibility genes in DEGs following DDX39B depletion. Resampling 100,000 times resulted in a distribution of the number of genes that overlap by chance, with 23 being the most common result. The observed overlap of 41 (Magenta arrow) demonstrates substantial enrichment (empirical p=0.00013). (**D**) Levels of DDX39B RNA and FOXP3 RNA, normalized to EEF1A1 RNA levels, after DDX39B depletion in CD4+ T cells and levels of DDX39B and FOXP3 protein relative to Tubulin after DDX39B depletion in CD4+ T cells. Connected dots indicate samples from the same donor. In all figures the error bars indicate standard deviation. *: p<0.05, **: p<0.01, ***: p<0.001 and ****: p<0.0001.

The online version of this article includes the following source data and figure supplement(s) for figure 1:

**Figure supplement 1.** Transcript level changes upon DDX39B depletion in primary human CD4+ T cells.

**Figure supplement 2.** Schematic diagram of the enrichment analysis of MS-susceptibility genes among genes whose expression is altered by DDX39B knockdown.

**Figure supplement 3.** Enrichment analysis of the MS_Pathogenic or MS_Protective genes.

**Figure supplement 4.** Coverage tracks of reads mapping to the *FOXP3* gene in control and DDX39B-depleted CD4+ T cells.

**Figure supplement 5.** Western blot analysis of FOXP3 expression upon DDX39B depletion in CD4+ T cells.

**Figure supplement 5—source data 1.** Two va files have been uploaded for all gels/blots shown in this figure: (1) the original files of the full raw unedited gels or blots and; (2) figures with the uncropped gels or blots with the relevant bands clearly labeled.

treatment (*Figure 1—figure supplement 1* and *Supplementary file 1*), which was consistent with Sh3 treatment reducing DDX39B RNA levels more profoundly (*Supplementary file 1*) and DDX39B protein levels more consistently (*Galarza-Muñoz et al., 2017*). Therefore, analysis of transcriptome alterations were first carried out by comparing CD4+ T cells depleted of DDX39B using Sh3 to cells treated with a control shRNA; however, all critical results were confirmed with both shRNAs and in samples from multiple human donors.

We evaluated the overlap between 762 transcripts that were significantly altered by Sh3-mediated DDX39B knockdown in CD4+ T cells from both donors 1 and 4 (*Figure 1A* and *Supplementary file 2*) and 558 putative MS susceptibility genes identified by the *Patsopoulos et al., 2019*. Expression of 421 of these MS susceptibility genes was detected in our RNAseq data and expression of 41 of these was significantly altered by DDX39B depletion (e.g. IL2RA, *Figure 1B*). This overlap meant that DDX39B differentially expressed genes (DEG) were highly enriched among MS susceptibility genes (p=0.00013; *Figure 1C* and *Figure 1—figure supplement 2*).

Based on eQTL information of the disease associated alleles (*GTEx Consortium, 2017*), we classified 250 of the 558 MS susceptibility genes as potentially 'pathogenic' since the disease-associated allele also associates with higher gene expression in lymphoblastoid cells, and 262 genes as potentially 'protective' since the disease-associated allele associates with lower expression of the gene (*Figure 1—figure supplement 3* and *Supplementary file 3*). We found significant overlap between MS pathogenic genes and genes upregulated upon DDX39B knockdown (p=0.00015, *Figure 1—figure supplement 3*), and also between MS protective genes and genes downregulated upon DDX39B knockdown, (p=0.03; *Figure 1—figure supplement 3*). In contrast, there was no significant overlap when the directional pairing was flipped to compare MS pathogenic genes with those downregulated by DDX39B knockdown (p=0.7), or MS protective genes with those upregulated by DDX39B knockdown (p=0.1; *Figure 1—figure supplement 3*). These data strongly suggested a shift of gene expression signature from MS-protective to MS-pathogenic upon DDX39B depletion, and provide support for a protective role for DDX39B in MS risk.

## DDX39B controls expression of FOXP3

Among the 762 transcripts that were significantly altered by Sh3-mediated DDX39B knockdown in CD4+ T cells from both donors (*Figure 1A*, *Supplementary file 2*) we identified Forkhead Box P3 (FOXP3) transcripts. Indeed, FOXP3 transcripts were among 122 high confidence targets of DDX39B significantly reduced in CD4+ T cells from both donors 1 and 4 with both Sh3 and Sh5 (*Figure 1—figure supplement 4*, *Supplementary file 1*). Given the importance of FOXP3 in the development, maintenance and function of Tregs (*Georgiev et al., 2019*; *Josefowicz et al., 2012*), and its strong association with autoimmune diseases (*Dominguez-Villar and Hafler, 2018*) we investigated this further.

Consistent with the measurement from the RNAseq, RT-qPCR showed FOXP3 RNA was reduced in DDX39B-depleted primary CD4+ T cells from the two aforementioned donors and from four additional healthy donors (*Figure 1D*). Importantly, FOXP3 protein was decreased in CD4+ T cells lysates that had measurable levels of total protein from four of these donors (*Figure 1D* and *Figure 1—figure supplement 5*).

In blood-derived CD4+ T cells FOXP3 is predominantly expressed in CD4+/CD25+ Tregs (*Josefowicz et al., 2012*), which are a small fraction of CD4+ T cells. To address the effect of DDX39B knockdown in Tregs we depleted DDX39B in the MT-2 Treg-like human cell line (*Hamano et al., 2015*). As observed in CD4+ T cells, knockdown of DDX39B led to markedly decreased expression of FOXP3 RNA and protein in MT-2 cells (*Figure 2A–C*). All FOXP3 isoforms detected in western blots were downregulated upon DDX39B knockdown (*Figure 2C*). Circulating primary human Tregs were isolated from PBMCs from Donors 7 and 8, or induced from PBMCs (*Liu et al., 2006*) from Donors 9 and 10 (iTregs), and depleted of DDX39B by expression of Sh3 and/or Sh5 (*Figure 2D*). DDX39B depletion led to a decrease in FOXP3 RNA in all Tregs from all four donors (*Figure 2D*) and a decrease in FOXP3 protein in Tregs from the two donors (7 and 8) where we had protein lysates (*Figure 2E*). All the above data indicate that robust FOXP3 expression in Tregs requires DDX39B.

## DDX39B depletion disrupts Treg-specific gene expression

Given that DDX39B knockdown led to low FOXP3 levels we predicted important alterations in immune relevant gene expression networks and tested this prediction using RNAseq data from primary CD4+ T cells depleted of DDX39B. Using Gene Set Enrichment Analysis (GSEA) (*Subramanian al., 2005*), we identified gene sets enriched in genes differentially expressed upon DDX39B knockdown in CD4+ T cells (*Supplementary file 4*). Among 648 gene sets enriched for genes downregulated upon DDX39B knockdown (nominal p-value ≤0.05) we found many related to immune function, including several gene sets of FOXP3 targets (*Gavin et al., 2007*; *Figure 3A* and *Figure 3—figure supplement*

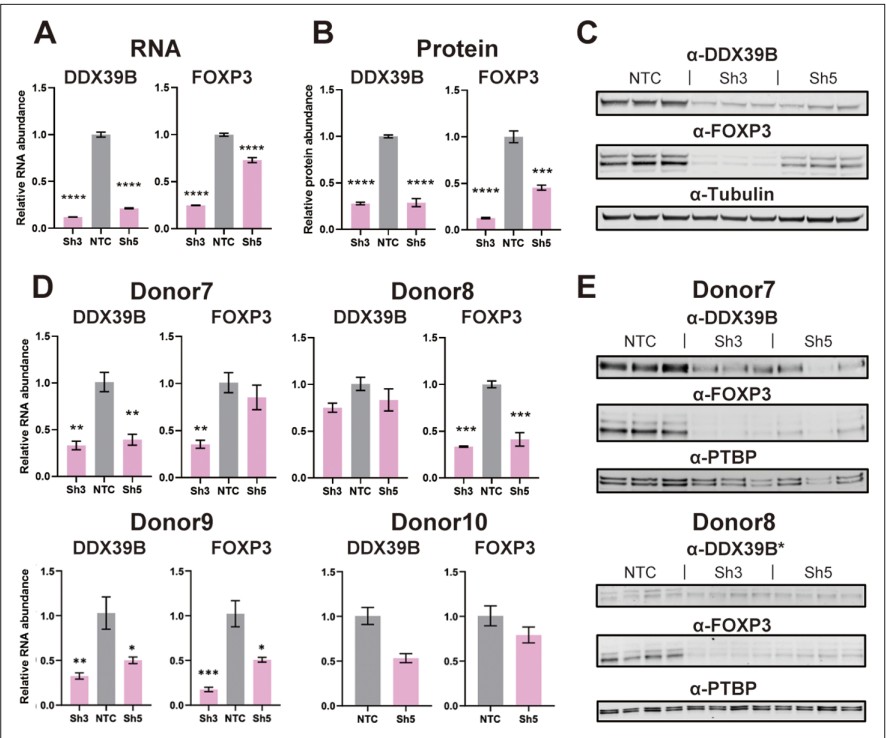

**Figure 2.** Effect of DDX39B depletion on FOXP3 RNA and protein expression. (**A**) Quantification of DDX39B and FOXP3 RNA between control (NTC) and DDX39B-depleted (Sh3 and Sh5) MT-2 cells. DDX39B and FOXP3 RNA abundance was measured by RT-qPCR and normalized to EEF1A1 RNA expression. (**B**) Quantification of DDX39B and FOXP3 protein expression between control (NTC) and DDX39B-depleted (Sh3 and Sh5) MT-2 cells from western blot in (**C**). DDX39B and FOXP3 protein abundance was normalized to alpha-tubulin. (**D**) DDX39B and FOXP3 expression upon DDX39B depletion in primary or induced Tregs. DDX39B was depleted in primary Tregs (Donors 7 and 8) and induced Tregs (Donors 9 and 10) via lentivirus transduction with either non-targeting control (NTC) or DDX39B targeting (Sh3 and/or Sh5) shRNAs. DDX39B and FOXP3 RNA abundance was measured by RT-qPCR and normalized to EEF1A1 RNA expression. (**E**) Western blot of DDX39B and FOXP3 expression between control (NTC) and DDX39B-depleted (Sh3 and Sh5) primary Tregs. PTBP was used as a loading control. Error bars indicate SEM except for NTC in Donor 10 where error bars represent range due to loss of one replicate sample; *: p<0.05, **: p<0.01, ***: p<0.001 and ****: p<0.0001.

The online version of this article includes the following source data for figure 2:

**Source data 1.** Two versions of source data files have been uploaded for all gels/blots shown in this figure: (1) the original files of the full raw unedited gels or blots and; (2) figures with the uncropped gels or blots with the relevant bands clearly labeled.

*1*). To confirm regulation of FOXP3-driven gene expression networks by DDX39B in an independent data set, we explored the ConnectivityMap (CMap) database provided by The Broad Institute (*Subramanian et al., 2017*). CMap is a large-scale catalog of transcriptional responses of human cells to chemical and genetic perturbations that enables the identification of conditions causing similar transcriptional responses. We found that the transcriptional profile of DDX39B knockdown (ID: CGS001-7919) showed significant overlap with that of FOXP3 knockdown (ID: CGS001-50943, *Supplementary file 5*). The GSEA and CMap results suggested that the gene expression networks of DDX39B and FOXP3 are tightly linked, which is consistent with robust FOXP3 expression requiring DDX39B.

We evaluated the effect of DDX39B knockdown on well-characterized FOXP3 targets, and established that some targets (e.g. IL2RA (CD25)) were significantly downregulated in DDX39B-depleted CD4[+] T cells and primary Tregs (*Figure 3B*). Other DDX39B targets behaved differently in these two cell populations, for instance EBI3, ICAM1, and TNFRSF9 were consistently downregulated by DDX39B depletion in CD4[+] T cells but not in Tregs (*Figure 3C*, *Figure 3—figure supplements 2–3*). These observations are consistent with the the existence of FOXP3-dependent and FOXP3-independent pathways altered by DDX39B knockdown.

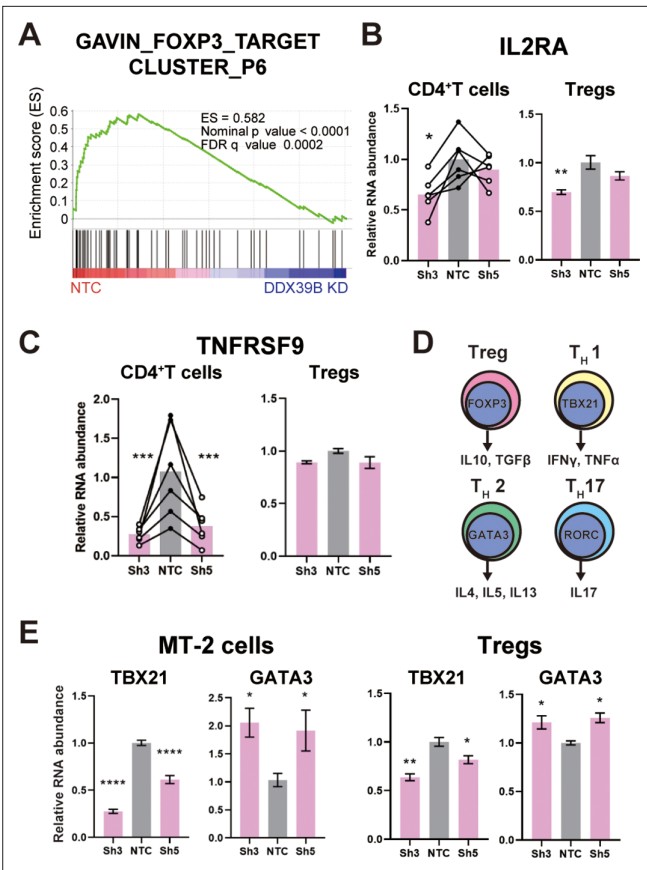

**Figure 3.** DDX39B depletion causes loss of Treg gene expression signature. (**A**) GSEA results of GAVING_FOXP3_TARGET_CLUSTER_P6 gene set enriched in normal over DDX39B depleted CD4[+] T cells (**B**) RNA abundance of IL2RA (CD25) relative to EEF1A1 in DDX39B-depleted CD4[+] T cells (Left) and primary Tregs (Right). (**C**) RNA abundance of TNFRSF9 relative to EEF1A1 in DDX39B-depleted CD4[+] T cells (Left) and primary Tregs (Right). (**D**) Schematic of genes expressed in different T cell lineages: Tregs, T$_H$17, T$_H$1, and T$_H$2 effector cells. (**E**) RNA abundance of transcription factors regulating T cell differentiation: TBX21 (T$_H$1), GATA3 (T$_H$2) in MT-2 cells (Left) and primary Tregs (Right). *: $p<0.05$, **: $p<0.01$, ***: $p<0.001$ and ****: $p<0.0001$.

The online version of this article includes the following figure supplement(s) for figure 3:

**Figure supplement 1.** GSEA results of FOXP3-related gene sets enriched in normal over DDX39B-depleted cells.

**Figure supplement 2.** Effect of DDX39B depletion on expression of genes in CD4[+] T cells.

**Figure supplement 3.** Effect of DDX39B depletion on expression of Treg genes.

We explored the effect of DDX39B knockdown on the expression of important transcriptional regulators of different T cell lineages (**Figure 3D**). DDX39B knockdown in MT-2 cells and primary Tregs decreased expression of TBX21, a transcriptional regulator of T$_H$1 cells, but increased expression of GATA3, a transcriptional regulator of T$_H$2 cells (**Figure 3E**). Given the complex composition of T cell populations (**Josefowicz et al., 2012**) these results have to be carefully interpreted; nonetheless, they suggest DDX39B depletion led to gene expression patterns associated with T$_H$2 cells.

## Low DDX39B exacerbates retention of *FOXP3* introns

We investigated the mechanistic basis for the dependence of FOXP3 RNA accumulation on DDX39B. Since DDX39B plays important roles in constitutive splicing (**Fleckner et al., 1997**; **Kistler and Guthrie, 2001**; **Shen et al., 2008**), alternative splicing (**Galarza-Muñoz et al., 2017**; **Nakata et al., 2017**), and nucleo-cytoplasmic transport (**Gatfield et al., 2001**; **Huang et al., 2018**; **Jensen et al., 2001**; **Luo et al., 2001**) we investigated which of these roles was affected in FOXP3 transcripts. We depleted DDX39B in the Treg-like MT-2 cells and biochemically fractionated cytoplasm, nucleoplasm and chromatin (**Figure 4A**) to determine which cellular compartment(s) showed reduced FOXP3 transcripts. In

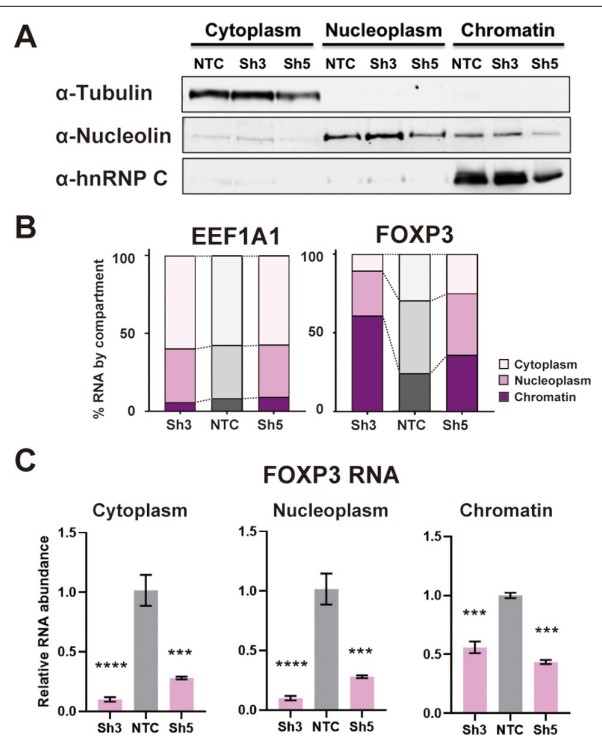

**Figure 4.** DDX39B depletion disturbs an early step in FOXP3 RNA biogenesis. (**A**) Protein abundance of subcellular compartment markers in fractionated control (NTC) or DDX39B-depleted (Sh3 and Sh5) MT-2 cells. (**B**) Percent EEF1A1 or FOXP3 RNA in subcellular compartments. (**C**) FOXP3 RNA abundance relative to EEF1A1 in subcellular compartments upon DDX39B depletion.

The online version of this article includes the following source data for figure 4:

**Source data 1.** Two versions of source data files have been uploaded for all gels/blots shown in this figure: (1) the original files of the full raw unedited gels or blots and; (2) figures with the uncropped gels or blots with the relevant bands clearly labelled.

control conditions, FOXP3 transcripts were disproportionally found to be associated with chromatin or in the nucleoplasm relative to transcripts coding for the translation elongation factor EEF1A1, and this association was increased for FOXP3 but not EEF1A1 RNAs upon DDX39B knockdown (**Figure 4B**). A significant reduction of FOXP3 RNAs was clearly observed in all three cellular fractions and the dramatic reduction in both nucleoplasm and cytoplasm argues against an effect on nucleocytoplasmic transport (**Figure 4C**). These data indicate that the effect of DDX39B knockdown on FOXP3 transcripts occurred early during their biogenesis.

Given this early effect, we analyzed changes in *FOXP3* RNA splicing. The RNAseq data suggested that DDX39B knockdown in CD4[+] T cells led to increased retention of *FOXP3* introns (**Figure 5A**). The DDX39B-sensitive retention of multiple *FOXP3* introns was confirmed using *FOXP3* intron-specific RT-qPCR from total primary CD4[+] T cell RNA (**Figure 5B**), total and chromatin-associated MT-2 RNA (**Figure 5C** and **Figure 5—figure supplement 1**), and total Treg RNA (**Figure 5—figure supplement 2**).

Retention of *FOXP3* introns 2, 4, 6, 7, 9, and 11 in DDX39B depleted MT-2 cells was rescued by expression of recombinant DDX39B (**Figure 5D and E**). FOXP3 protein expression was also rescued although to a lesser extent than intron retention (**Figure 5D**). The lack of strong rescue of FOXP3 protein expression is likely due to transcriptional downregulation of *FOXP3* secondary to the sustained knockdown of FOXP3 preceeding the expression of DDX39B. Nonetheless, given the robust rescue of the splicing phenotype we concluded that DDX39B is required for efficient intron removal for several *FOXP3* introns and this results in a dramatic decrease in overall levels of *FOXP3* RNA and protein.

To ascertain if changes in intron retention were seen for other DDX39B targets, we examined global changes in alternative splicing upon DDX39B knockdown in our CD4[+] T cell RNAseq data.

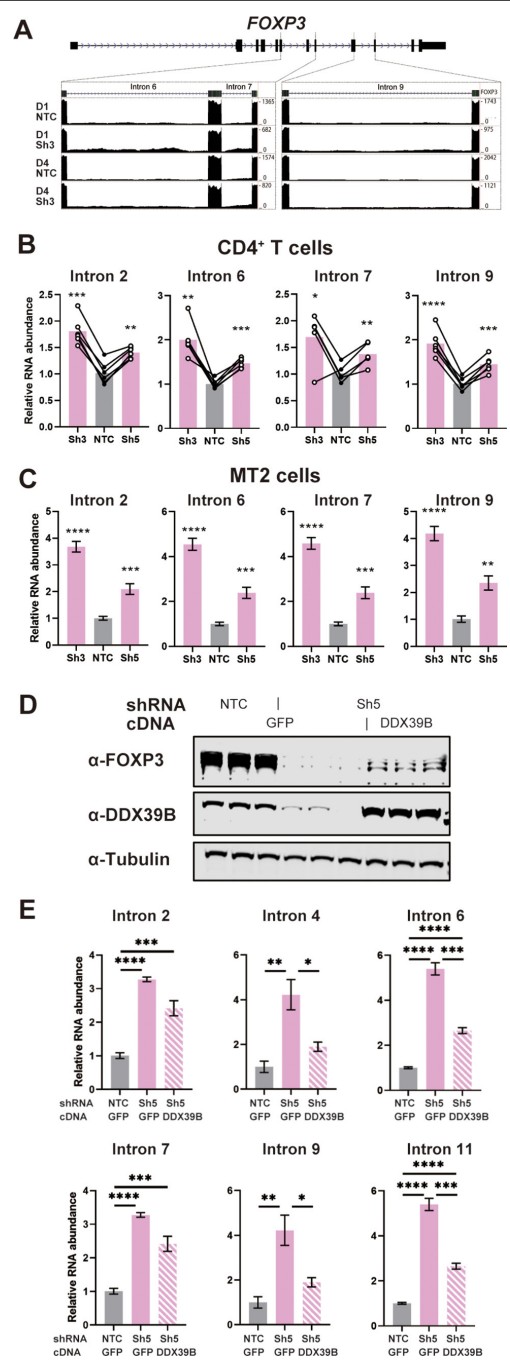

**Figure 5.** DDX39B depletion triggers the retention of FOXP3 introns. (**A**) RNAseq reads mapping to the FOXP3 genomic region in control (NTC) or DDX39B depleted (Sh3) CD4⁺ T cells from Donors 1 and 4. Read counts for *FOXP3* introns 6, 7, and 9 are shown on the Y-axis in the two insets. (**B, C**) Abundance of FOXP3 RNA introns relative to total FOXP3 RNA after DDX39B depletion in CD4⁺ T cells from six donors (**B**) or MT-2 cells (**C**). (**D–E**) Rescue of the DDX39B depletion (Sh3) by exogenous expression (GFP or DDX39B) in MT-2 cells. The abundance of FOXP3 and DDX39B relative to Tubulin (**D**) or FOXP3 RNA introns relative to total

*Figure 5 continued on next page*

*Figure 5 continued*

FOXP3 RNA (**E**) are shown.\*: p<0.05, \*\*: p<0.01, \*\*\*: p<0.001 and \*\*\*\*: p<0.0001.

The online version of this article includes the following source data and figure supplement(s) for figure 5:

**Source data 1.** Two versions of source data files have been uploaded for all gels/blots shown in this figure: (1) the original files of the full raw unedited gels or blots and; (2) figures with the uncropped gels or blots with the relevant bands clearly labeled.

**Figure supplement 1.** Quantification of *FOXP3* retained introns in the chromatin fraction of MT-2 cells.

**Figure supplement 2.** Quantification of *FOXP3* retained introns in primary and induced Tregs.

**Figure supplement 3.** Alternative splicing events observed upon DDX39B depletion in CD4⁺ T cells.

First and foremost, we noted that only a limited number of splicing events detected by RNAseq were significantly altered by DDX39B depletion (*Figure 5—figure supplement 3*). As with transcript level changes, we noted more alternative splicing events changing with Sh3 than Sh5, and more with Donor 1 than Donor 4 (*Figure 5—figure supplement 3*). We detected very few events in common for all conditions, perhaps because of the inefficient knockdown of DDX39B observed in the libraries from cells treated with Sh5 and the well-documented heterogeneity of alternative splicing between individuals (*Wang et al., 2008*). Therefore, most of our analysis focused on events changed by Sh3 in CD4⁺ T cells from Donors 1 and 4 (*Supplementary file 6*). Retained introns were the most frequently observed type of RNA splicing event changed (n=784) with increased retention in 397 introns and decreased retention in 387 (*Figure 5—figure supplement 3*). The second most common changes were in cassette exon use (n=488), followed by alternative use of 5' and 3' splice sites (*Figure 5—figure supplement 3*). We conclude that splicing of a subset of genes is sensitive to DDX39B levels and this is primarily due to modulation of intron retention as observed for *FOXP3* transcripts.

## *FOXP3* introns have C-rich py tracts

Since removal of *FOXP3* introns was sensitive to DDX39B levels and since *FOXP3* is on the X chromosome, we compared *FOXP3* splice sites to those found in introns of other protein coding genes on the X chromosome. 5' splice sites in *FOXP3* introns were only modestly weaker than those found in the average gene, whereas 3'

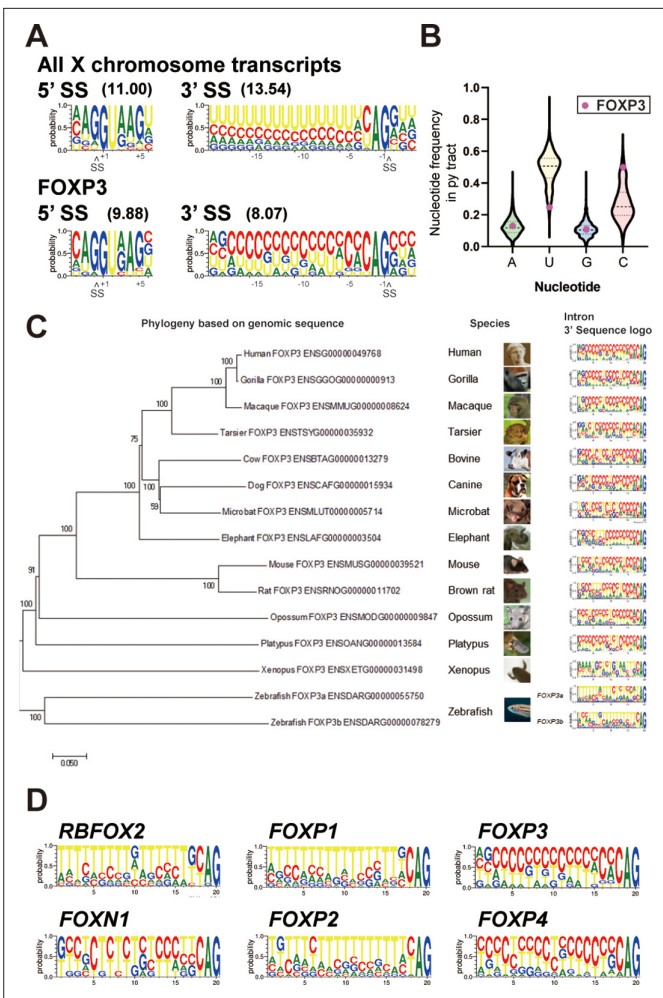

**Figure 6.** *FOXP3* introns have conserved C-rich py tracts. (**A**) Sequence logos of the 5' and 3' splice sites (SS) of *FOXP3* versus all other chromosome X genes. 5' or 3' MaxEntScore based on the highest-probability sequence are shown. (**B**) Distribution of the py tract nucleotide frequency among chromosome X genes. Magenta dots indicate *FOXP3*. (**C**) Phylogenic tree and 3' SS sequence logos of the *FOXP3* among 14 species. Zebrafish has two *FOXP3* genes (*FOXP3a* and *FOXP3b*). (**D**) 3' SS sequence logos of FOX family genes. *RBFOX2* is shown as an outgroup.

splice sites were considerably weaker (*Figure 6A*). This correlated with an altered composition of the polypyrimidine tract (py tract): *FOXP3* introns contain uracil (U)-poor and cytosine (C)-rich py tracts, which on the average are more likely to have a C at each position, except positions −19 and −20 (measured relative to the guanine (G) in the 3' splice site AG marked as position −1) (*Figure 6A and B* and *Supplementary file 7*). This contrasts with our analysis of py tracts in other genes on the X chromosome (*Figure 6A and B*) and previous analysis for all human introns spliced by the major (U2 snRNP dependent) spliceosome (*Yeo and Burge, 2004*), where the average position in the py tract is most likely to be occupied by U.

C-rich py tracts in *FOXP3* introns are conserved in mammals from both monotreme and theriiformes subclasses, which diverged over 200 million years ago (*Tarver et al., 2016*; *Figure 6C* and *Supplementary file 7*). Furthermore, the amphibian *Xenopus laevis FOXP3* has a U-poor py tract, although in this case the Us are replaced by all three other nucleotides (*Figure 6C*). The zebrafish *Danio rerio* has two *FOXP3* paralogs and py tracts in one of these (*FOXP3a*) trend towards the C-rich py tracts of mammals, while the other (*FOXP3b*) has U-rich tracts (*Figure 6C*). An examination of other FOX family genes revealed that close relatives of *FOXP3* diverge in their py tracts, *FOXP4* shares C-rich py tracts, while *FOXP1* and *FOXP2* have U-rich tracts (*Figure 6D*). *RBFOX2*, which encodes for a RNA binding protein and not a member of the FOX family of transcription factors, is shown as a phylogenetic outlier with U-rich py tracts (*Figure 6D*). U-poor and C-rich py tracts appear to be

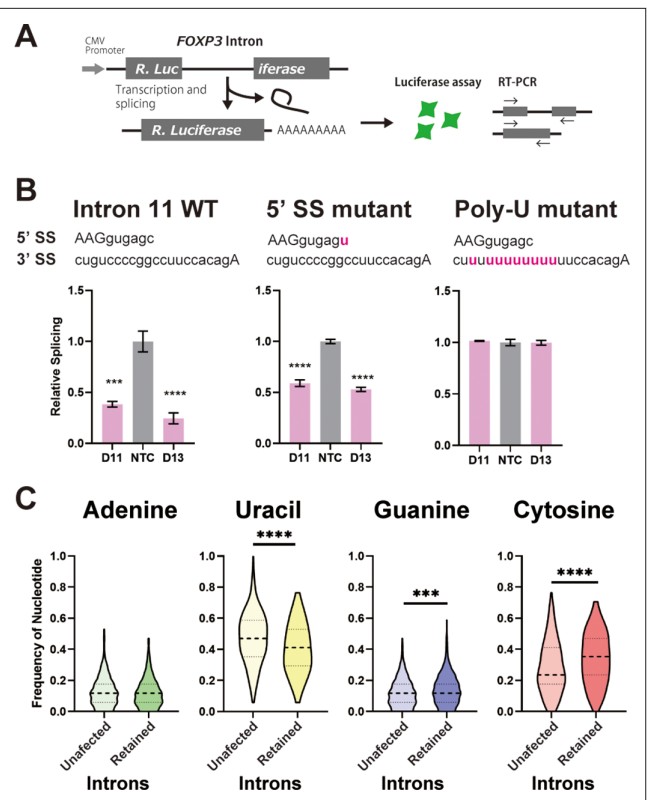

**Figure 7.** C-rich py tracts in *FOXP3* introns determine DDX39B sensitivity. (**A**) Schematic diagram of the splicing reporter. (**B**) Relative splicing efficiency of the wild-type (WT) or mutant (5' SS mutant and Poly-U mutant) FOXP3 intron 11 splicing reporters after DDX39B depletion with D11 and D13 siRNAs. The sequence of 5' splice sites (5' SS) for these introns is shown indicating the three upstream exonic residues in capital letters and the six downstream intronic residues in minuscule letters. The sequence of the py tracts and 3' splice sites (3' SS) for these introns is shown indicating the intronic residues in minuscule letters and a single exonic residue in capital letters. (**C**) Nucleotide frequency in the py tract of introns that are insensitive (unaffected) or sensitive (retained) to DDX39B depletion. *: $p < 0.05$, **: $p < 0.01$, ***: $p < 0.001$ and ****: $p < 0.0001$.

The online version of this article includes the following source data and figure supplement(s) for figure 7:

**Figure supplement 1.** Luciferase activity of the splicing reporters.

**Figure supplement 2.** RT-PCR analysis of splicing efficiency of wild-type or mutant *FOXP3* Intron 11 RLuc reporters.

**Figure supplement 2—source data 1.** Two versions of source data files have been uploaded for all gels/blots shown in this figure: (1) the original files of the full raw unedited gels or blots and; (2) figures with the uncropped gels or blots with the relevant bands clearly labeled.

**Figure supplement 3.** RT-PCR analysis of splicing efficiency of wild-type or mutant *FOXP3* Intron 7 RLuc reporters.

**Figure supplement 3—source data 1.** Two versions of source data files have been uploaded for all gels/blots shown in this figure: (1) the original files of the full raw unedited gels or blots and; (2) figures with the uncropped gels or blots with the relevant bands clearly labeled.

**Figure supplement 4.** Comparison of Max entropy splice sites (SS) score between unaffected and DDX39B-sensitive introns.

**Figure supplement 5.** Comparison of Max entropy splice site (SS) scores (top panel) and nucleotide composition of py tracts (bottom panel) of unaffected introns or introns that were more retained in CD4[+] T cells from either donor 1 or 4 depleted of DDX39B by treatment with Sh3 or Sh5.

**Figure supplement 6.** Detection of intron retention events in X chromosome genes with C-rich or U-rich py tracts.

**Figure supplement 7.** Retention of *FOXP1* introns is not increased upon DDX39B depletion in MT-2 cells.

**Figure supplement 8.** RT-PCR analysis of splicing efficiency of wild-type or mutant *FOXP1* Intron 19 RLuc reporters.

*Figure 7 continued on next page*

*Figure 7 continued*

**Figure supplement 8—source data 1.** Two versions of source data files have been uploaded for all gels/blots shown in this figure: (1) the original files of the full raw unedited gels or blots and; (2) figures with the uncropped gels or blots with the relevant bands clearly labeled.

**Figure supplement 9.** Comparison of py tract composition between unaffected and less-retained introns.

the ancestral form in the *FOX* family as py tracts in the more divergent *FOXN1* are more like those in *FOXP3* and *FOXP4* (*Figure 6D*). The evolutionary conservation of C-rich py tracts in *FOXP3* and other *FOX* family genes suggests important regulatory function.

## C-rich py tracts are required for the sensitivity of *FOXP3* introns to DDX39B depletion

Since we had previously shown that replacement of U with C in intronic py tracts inhibits splicing of model pre-mRNA substrates in vitro (*Roscigno et al., 1993*), we posited that the C-rich py tracts in *FOXP3* introns would make these inefficient and highly sensitive to DDX39B levels. To test this, we made reporter constructs where *FOXP3* introns interrupted a Renilla luciferase ORF, and splicing efficiency could be inferred by luciferase activity or measured directly using RT-PCR (*Figure 7A*). Importantly, splicing reporters containing *FOXP3* introns were markedly dependent on DDX39B (*Figure 7B* and *Figure 7—figure supplements 1–3*). While conversion of the 5' splice site in *FOXP3* introns 7 and 11 to consensus only modestly relieved DDX39B dependency, replacement of their py tracts with U-rich tracts made these introns insensitive to DDX39B knockdown (*Figure 7B* and *Figure 7—figure supplements 2–3*). These findings indicated that weak C-rich py tracts are necessary for the strong DDX39B dependency of *FOXP3* introns.

To determine whether or not other DDX39B-sensitive intron retention events shared sequence features with *FOXP3* introns, we analyzed our RNAseq data from DDX39B-depleted CD4+ T cells. When 500 randomly selected unaffected introns were compared to 397 introns with increased retention upon DDX39B knockdown with Sh3 in either donor 1 or 4 we found no difference in the 5' splice site maximum entropy score and a modest but statistically significant decrease in the 3' splice site maximum entropy score (*Figure 7—figure supplement 4*). Since the lower 3' splice site scores for

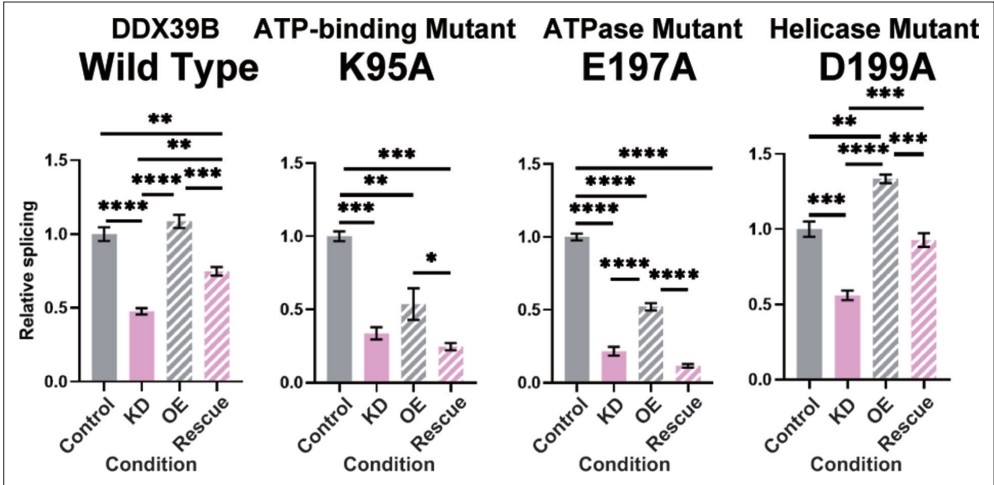

**Figure 8.** *FOXP3* intron 11 splicing requires DDX39B ATPase activity but not its helicase activity. Rescue of the splicing of FOXP3 intron 11 reporter by induced expression of wild type (WT) or mutant (K95A, E197A, and D199A) DDX39B. KD and OE indicate knockdown and over-expression, respectively. *: p<0.05, **: p<0.01, ***: p<0.001 and ****: p<0.0001.

The online version of this article includes the following source data and figure supplement(s) for figure 8:

**Figure supplement 1.** Expression of WT or mutant DDX39B in stable cell lines.

**Figure supplement 1—source data 1.** Two versions of source data files have been uploaded for all gels/blots shown in this figure: (1) the original files of the full raw unedited gels or blots and; (2) figures with the uncropped gels or blots with the relevant bands clearly labeled.

*FOXP3* introns were driven by the C-rich py tract, we examined the composition of the py tract in DDX39B-sensitive introns. We found that the frequency of U residues in the py tracts of DDX39B sensitive introns was significantly lower than in unaffected ones, and the frequency of C was significantly higher (*Figure 7C*). There was a statistically significant but very small increase in G residues in these U-poor, C-rich py tracts (*Figure 7C*). An equivalent analysis interrogating all intron retention events altered with DDX39B knockdown with both Sh3 and Sh5 led to same conclusion: DDX39B sensitive introns are more likely to have U-poor, C-rich py tracts than unaffected introns (*Figure 7—figure supplement 5*).

To further test our conclusions on the role of C-rich py tracts, we interrogated the behavior of introns in other X chromosome genes selected only because they contained either C-rich or U-rich py tracts. Of twelve introns with C-rich py tracts, in *FAM3A*, *PORCN*, *RBM10*, *RENBP*, *CFP*, and *G6PD*, all but two, both in the *CFP* gene, were more retained upon DDX39B knockdown in MT-2 cells (*Figure 7—figure supplement 6*). In contrast, four U-rich py tract introns in *FMR1* and *DDX3X* were tested and were not more retained upon DDX39B depletion (*Figure 7—figure supplement 6*), thereby supporting the requirement of C-rich py tracts for DDX39B-dependency.

The results above on *CFP* introns suggested that a U-poor, C-rich py tract was not sufficient to confer DDX39B sensitivity. To address this, we analyzed splicing of introns 11, 14, and 19 in the *FOXP1* gene (*Figure 7—figure supplement 7*), which encodes a homologue of *FOXP3* that contains introns with U-rich py tracts (*Figure 6D* and *Figure 7—figure supplement 7*). Retention of *FOXP1* introns 11, 14, and 19 was not increased with DDX39B knockdown in MT-2 cells (*Figure 7—figure supplement 7*; contrast to *FOXP3* introns in *Figure 5C*). We modified the *FOXP3* intron 11 luciferase reporter by replacing the intron with the first and last 75 nucleotides of *FOXP1* intron 19. As expected the *FOXP1* intron 19 reporter was not sensitive to DDX39B knockdown (*Figure 7—figure supplement 8*). We replaced the *FOXP1* intron 19 py tract with that of *FOXP3* intron 11 and noted that this C-rich tract did not confer DDX39B sensitivity on *FOXP1* intron 19 (*Figure 7—figure supplement 8*) indicating that the C-rich tract in *FOXP3* intron 11 was not sufficient for this sensitivity. Consistent with this, *en masse* analysis of the 387 introns that were less retained upon DDX39B depletion revealed a modest increase in C frequency in py tracts suggesting that this is not sufficient to confer DDX39B-dependency (*Figure 7—figure supplement 9*). Therefore, a C-rich py tract was required but not sufficient to impart DDX39B-dependency on an intron.

Collectively, the data presented in this section show that C-rich py tracts are required for the strong DDX39B-dependency of *FOXP3* introns and suggest the existence of a type of introns where the C-rich py tract is required for said dependency.

## Splicing of FOXP3 introns requires the ATPase activity of DDX39B but not its helicase activity

The py tract is recognized early in the splicing reaction by the U2 small nuclear RNA auxiliary factor (U2AF) composed of U2AF1 (U2AF[35]) and U2AF2 (U2AF[65]) (*Ruskin et al., 1988*; *Zamore and Green, 1989*). U2AF2 binds preferentially to U-rich sequences in the py tract in vitro (*Singh et al., 1995*; *Zamore et al., 1992*) and this is recapitulated by crosslinking-immunoprecipitation experiments in vivo (*Wu and Fu, 2015*). U2AF2 binds DDX39B, which licenses the U2AF-mediated recruitment of U2 snRNP to the neighboring branchpoint sequence to form the pre-spliceosome (*Shen et al., 2008*). We conjectured that the exquisite dependency of C-rich py tract FOXP3 introns on DDX39B was mediated by its ability to promote pre-spliceosome formation. Pre-spliceosome formation requires the ATPase activity of DDX39B, but not its helicase activity, which is required for later steps in spliceosome assembly (*Shen et al., 2008*). While expression of wildtype (WT) DDX39B rescued splicing of FOXP3 intron 11 in DDX39B depleted cells, two ATPase defective mutants, K95A and E197A (*Shen et al., 2007*), did not rescue splicing (*Figure 8* and *Figure 8—figure supplement 1*). On the other hand, the helicase defective DDX39B mutant D199A that retains ATPase activity and supports pre-spliceosome formation (*Shen et al., 2008*; *Shen et al., 2007*) rescued FOXP3 intron 11 splicing (*Figure 8* and *Figure 8—figure supplement 1*). Of note, the data in *Figure 8* indicate that the DDX39B D199A mutant rescued FOXP3 intron 11 splicing even better than the WT DDX39B. This result could be explained by the fact that while the D199A lacks helicase activity, it has higher ATPase activity than WT DDX39B (*Shen et al., 2007*). These results indicate that the C-rich py tract of FOXP3 intron 11 is highly dependent on the DDX39B ATPase activity and this unique dependency may distinguish its

spliceosome assembly pathway from more conventional introns. Nine of eleven FOXP3 introns belong to the C-rich py tract type, and are all likely to share this requirement for the DDX39B ATPase activity, which explains why overall FOXP3 expression is exquisitely sensitive to DDX39B levels.

## Discussion

DDX39B (also known as BAT1) was proposed to be an anti-inflammatory factor (*Allcock et al., 2001*) and we have shown that *DDX39B* is genetically associated with MS, an autoimmune disease (*Galarza-Muñoz et al., 2017*). Previously we discovered a molecular mechanism that partially explained these phenomena. DDX39B is a potent activator of IL7R exon 6 splicing and a repressor of the production of sIL7R (*Galarza-Muñoz et al., 2017*), which exacerbates the MS-like EAE (*Lundström et al., 2013*). Complementing this molecular analysis, we demonstrated epistasis between alleles in DDX39B and IL7R and MS risk, with individuals homozygous for the risk alleles at both loci having an approximately 3-fold higher chance of developing MS (*Galarza-Muñoz et al., 2017*). While this connection between DDX39B and autoimmunity was compelling, evolutionary arguments suggested that this RNA helicase would play other roles in immunity. Our exploration of this suggestion led to the discovery, described above, that DDX39B is exquisitely required for the splicing of FOXP3 transcripts. Low levels of DDX39B lead to low levels of FOXP3 and by inference low numbers of functional Tregs.

Our work highlights the importance of post-trancriptional mechanisms in the regulation of FOXP3. These have not been noted before and indeed CRISPR screens to discover regulators of FOXP3 did not discover DDX39B (*Cortez et al., 2020*; *Loo et al., 2020*; *Schumann et al., 2020*), but this is likely because knockout of DDX39B results in lethality.

The connection between FOXP3 and autoimmunity is well established: mutations in FOXP3 cause a systemic and severe autoimmune syndrome in humans called immunodysregulation polyendocrinopathy enteropathy X-linked (IPEX) syndrome (*Bennett et al., 2001*; *Chatila et al., 2000*; *Wildin et al., 2001*) and the equivalent scurfy in mice (*Brunkow et al., 2001*). While there is evidence supporting a prominent role for FOXP3 in MS (*Fletcher et al., 2009*; *Sambucci et al., 2018*), genetic variation within FOXP3 has yet to be associated with MS in large genome wide associations studies largely due to the fact that the X chromosome has been understudied in human genetic association studies. The role of Tregs, which depend on FOXP3 transcriptional control (*Hori et al., 2003*), on autoimmunity has been carefully documented (*Dominguez-Villar and Hafler, 2018*). Furthermore, it is clear that high expression of FOXP3 is required for full Treg function and there is a FOXP3 dose-dependent effect on immune supressive activity (*Wan and Flavell, 2007*). Therefore, given the strong dependence of FOXP3 expression on DDX39B levels, it is likely that humans with low levels of DDX39B will have reduced FOXP3 expression and by extension low levels of Treg cell development, maintenance and suppressive function. Furthermore, given our data that DDX39B knockdown promotes GATA3 expression it is possible that low levels of DDX39B could promote the conversion of Tregs to $T_H2$-like self-reactive effector cells (*Komatsu et al., 2014*; *Noval Rivas et al., 2015*; *Zhou et al., 2009*). The DDX39B control of FOXP3 expression cements its role in immune regulation.

There are other intriguing connections between DDX39B and immunity. First, we show that putative MS susceptibility genes are regulated by DDX39B and some are likely regulated independently of effects on FOXP3. Second, DDX39B regulates the nucleocytoplasmic transport of human circular RNAs (circRNAs) larger than 1200 nucleotides (*Huang et al., 2018*). There is an emerging realization that circRNAs play important, although heretofore incompletely understood roles, in immunity and autoimmune disease (*Zhou et al., 2019*). Third, *DDX39B* resides in the class III region of the MHC and many of its neighbors code for proteins involved in RNA transactions (e.g. *DXO* and *SKIV2L*) (*Lehner et al., 2004*; *Schott and Garcia-Blanco, 2021*). This overrepresentation of genes involved in RNA metabolism within the class III region is found in all vertebrates we have examined, suggesting an important immune function for these RNA binding proteins. These observations, first revealed by studying DDX39B, suggest new and interesting connections between RNA and immunity.

DDX39B, which is also known as UAP56, was characterized as a factor required for formation of splicing complexes (*Fleckner et al., 1997*; *Shen et al., 2008*). Here we describe a type of introns that have C-rich py tracts and depend on high levels of DDX39B for splicing. Although these py tracts are poor binding sites for U2AF2 (*Singh et al., 1995*), we propose that DDX39B-driven U2 snRNP binding to the branchpoint sequence stabilizes U2AF2 at these suboptimal sites. We suggest that for these introns the U2 snRNP complex may be the first splicing commitment complex, which

would be distinct from U2 snRNP-independent commitment complexes in introns with U-rich py tracts (*Jamison et al., 1992*; *Li et al., 2019*; *Seraphin and Rosbash, 1989*). Our studies indicate that these introns are exceedingly sensitive to DDX39B ATPase activity, but much less so to its helicase activity. Shen et al showed that DDX39B ATPase activity, but not helicase activity, is required for binding of U2AF and recruitment of U2 snRNP to the branchpoint sequence of the 3' splice site to form the pre-spliceosome (*Shen et al., 2008*). This suggests that once bound to the branchpoint sequence and py tract in these C-rich introns, the U2 snRNP-U2AF complex does not require further remodeling by DDX39B. In contrast to this, activation of IL7R exon 6 requires DDX39B helicase activity (*Galarza-Muñoz et al., 2017*). Our data thus indicate distinct functions for DDX39B in different splicing events, which implies multiple pathways to spliceosome assembly as has been suggested before (*Kistler and Guthrie, 2001*; *Newnham and Query, 2001*). An intriguing possibility is that FOXP3 introns, and perhaps many introns with C-rich py tracts, are actually prone to intron 'detention' rather than retention (*Boutz et al., 2015*) and that certain stimuli may free them from DDX39B dependency.

## Materials and methods

### Cells

MT-2 cells were kindly provided by Bryan R. Cullen (Duke University) and grown at 37 °C in Roswell Park Memorial Institute (RPMI) 1640 medium (Thermo Fisher Scientific) with 10% (v/v) fetal bovine serum (FBS) (Genesee Scientific) and 1% (v/v) penicillin-streptomycin (Thermo Fisher Scientific). Human embryonic kidney 293T/17 (HEK 293T/17; ATCC CRL-11268) and HeLa (ATCC CCL-2) cell lines were obtained from the Duke University Cell Culture Facility. The HeLa Flp-In T-Rex cell line (HeLa-Flp-In) was kindly provided by Dr. E. Dobrikova (Duke University). HEK293T/17, HeLa, and HeLa-Flp-In cells were cultured at 37 °C in Dulbecco's Modified Eagle Medium (DMEM) (Thermo Fisher Scientific) with 10% (v/v) FBS, 1% (v/v) penicillin-streptomycin and 2.5 μg/ml Plasmocin (InvivoGen). HeLa_Flp-In cell lines expressing wild-type or mutant versions of DDX39B (K95A, E197A and D199A) were grown as described above except media was supplemented with 2.5 μg/ml blasticidin (InvivoGen) and 200 μg/ml Hygromycin B (Thermo Fisher Scientific). All cells were free of Mycoplasma contamination as confirmed by routine testing using the MycoAlert Mycoplasma Detection Kit (Lonza).

Primary human CD4$^+$ T cells were isolated and cultured as described previously (*Galarza-Muñoz et al., 2017*). In brief, peripheral blood mononuclear cells (PBMCs) from healthy donors were isolated from whole blood or buffy coats using the Ficoll-Plaque Gradient method. Blood samples were collected at Duke University following the institutional IRB protocol (# Pro00070584) and buffy coats were purchased from the New York Blood Center. CD4$^+$ T cells were further isolated using CD4$^+$ T cell isolation kit (Miltenyi Biotec). The isolated cells were cultured in RPMI medium supplemented with 20% (v/v) FBS and 100 ng/mL human recombinant IL-2 (Peprotech). Two days prior to transduction, CD4$^+$ T cells or MT-2 cells were activated with media containing 50 ng/mL anti-CD3 (eBioscience) and 100 ng/mL anti-CD28 (BD Biosciences) antibodies.

Primary human T regulatory cells (Tregs) were isolated from PBMCs using CD4$^+$ CD25$^+$ CD127$^{dim/-}$ Regulatory T Cell Isolation Kit (Miltenyi Biotec). The isolated Treg cells were expanded in culture for 14 days with TexMACS medium (Miltenyi Biotec) supplemented with 5% (v/v) Human AB Serum (Sigma), 500 IU/mL recombinant human IL2 (PeproTech) and Treg Expansion kit (Miltenyi Biotec) which contains beads pre-loaded with anti-CD3 and anti-CD28 antibodies. Induced T regulatory cells (iTregs) were purchased from IQ Biosciences (IQB-Hu1-iTr-1) and were expanded in culture for 6 days using the same media and Treg Expansion kit as the primary human Tregs.

### Plasmids

shRNA pLKO.1 plasmids expressing non-targeting control shRNA (NTC: SHC002) or anti-DDX39B shRNAs (Sh3: TRCN0000286976; and Sh5: TRCN0000294383) were purchased from Millipore Sigma. The construct pLCE-DDX39B was generated by cloning the coding sequence of *DDX39B* into the pLCE vector. Splicing reporter plasmids (pcDNA3.1-RLuc_FOXP3 Intron 7, pcDNA3.1-Rluc_FOXP3 Intron 9, pcDNA3.1-Rluc_FOXP3 Intron 11, and pcDNA3.1-Rluc_HGB1 Intron 2) were constructed by inserting introns 7, 9 or 11 of FOXP3 (NC_000023), or intron 2 of HGB1 (NC_000011) within the open reading frame of *Renilla Luciferase* (Rluc) at nucleotide positions 599, 601, 408, or 253, respectively, in the pcDNA3.1 vector (pcDNA3.1-Rluc). Splice sites (SS) mutations were introduced in the

corresponding parental plasmid using In-Fusion HD Cloning kit (Takara Bio) to generate pcDNA3.1-Rluc_FOXP3 Intron 7 5'-SS G-U, pcDNA3.1-Rluc_FOXP3 Intron 7 3'-SS PolyU, pcDNA3.1-Rluc_FOXP3 Intron 11 5'-SS G-U, and pcDNA3.1-Rluc_FOXP3 Intron 11 3'-SS PolyU. To construct pcDNA3.1-Rluc_FOXP1 Intron 19 WT and 3' C-rich py tract mutant, synthesized DNA fragments with the FOXP1 intron sequences (NG_028243.1) were purchased from Integrated DNA Technologies and were subcloned into the pcDNA3.1-Rluc plasmid. Point mutants DDX39B K95A, E197A and D199A were introduced in the parental pcDNA5/FRT/TO-DDX39B plasmid (*Galarza-Muñoz et al., 2017*) using QuickChange Lightning Mutagenesis Kit (Agilent Technologies). All constructs and mutations were confirmed by Sanger sequencing. Primers used to generate these constructs are shown in *Supplementary file 8*.

## Establishment of stable cell lines

Generation of the inducible HeLa cell line expressing wild-type DDX39B or DDX39B mutant D199A was described previously (*Galarza-Muñoz et al., 2017*). HeLa cell lines stably expressing DDX39B mutants (K95A and E197A) were constructed similarly. In brief, HeLa Flp-In T-Rex cells were co-transfected with these constructs in the pcDNA5/FRT/TO plasmid and pOG44 plasmid, which encodes Flp recombinase, using Lipofectamine 2000 (Thermo Fisher Scientific). Transfected cells were selected with 2.5 µg/ml blasticidin and 200 µg/ml Hygromycin B for 15 days. Expression of the transgene was induced by addition of 1 µg/mL doxycycline in the culturing media.

## Lentiviral packaging

Lentiviral packaging of shRNA constructs was conducted in 293T/17 cells using Lipofectamine 2000 (Thermo Fisher Scientific). In brief, $1.0 \times 10^7$ 293 T/17 cells were seeded in 15 cm dishes in DMEM media and cultured overnight (three 15 cm dishes per construct). Cells were co-transfected with 17 µg of the corresponding shRNA pLKO.1 vector, pLCE-DDX39B or pLCE-GFP, 17 µg of packaging plasmid (pCMVR8.74) and 7 µg of VSV-G envelope plasmid (pMD2.G) in serum-free media and 160 µL of Lipofectamine 2000. The media was replaced with 20 mL fresh DMEM media 18 hr post-transfection. After 72 hr, supernatants were collected, filtered through 0.45 mm filters and concentrated to 6 mL in Amicon Ultra 100 K centrifugal filter units (Millipore Sigma). Concentrated lentiviral particles were used immediately or stored at –80 °C.

## Lentiviral transduction of human primary CD4+ T cells, MT-2 cells and primary Tregs

Transduction of human primary CD4+ T cells with either NTC or anti-DDX39B shRNAs was described previously.(*Galarza-Muñoz et al., 2017*) In brief, $4.0 \times 10^6$ activated primary CD4+ T cells from each donor were transduced in T25 flasks with lentiviruses encoding control (NTC) or DDX39B-targeting (Sh3 or Sh5) shRNAs for 3 days, and transduced cells were selected under medium supplemented with 1.5 µg/mL puromycin for 4 days. Cells were then cultured for 24 hr in the absence of puromycin and collected for functional analyses (8 days after initial transduction). Transduction of activated MT-2 cells ($1.0 \times 10^7$ cells) was carried out as for primary CD4+ T cells. Activated primary Tregs ($1.0 \times 10^5$ cells) were transduced in 96-well plates and cultured for 7 days with half media exchange every 2 days (without puromycin selection). Induced Tregs ($1.0 \times 10^5$ cells) were transduced in 96 well plates with Sh3 and/or Sh5 for two days then subjected to puromycin selection for two days. Cells were then cultured for an additional 24 hr in the absence of puromycin and harvested for functional analyses. In all cell types, depletion of DDX39B protein was confirmed by western blot using anti-UAP56 (DDX39B) antibody (ab18106, Abcam) and α-Tubulin (AB_1904178, Cell Signaling Technology), PTBP1 (anti-PTB rabbit serum *Wagner and Garcia-Blanco, 2002*), or α-beta-actin (sc-47778, Santa Cruz) as loading control.

## DDX39B rescue in MT-2 cells

To rescue DDX39B expression in MT-2 cells, $1.0 \times 10^7$ cells were co-transduced with the following combination of shRNAs and expression plasmids for 3 days: (i) NTC shRNA +pLCE GFP (control), (ii) Sh5 shRNA +pLCE GFP (knockdown), and (iii) Sh5 shRNA +pLCE-DDX39B (rescue). The cells were then selected with RPMI medium supplemented with 1.5 µg/mL puromycin for 2 days, followed by 24 hr culture in the absence of puromycin, and collection of cell lysates for RNA and protein analyses (6 days after initial transduction). We used the DDX39B shRNA Sh5 since it targets the 3' untranslated region of DDX39B mRNA, and thus it depletes the endogenous DDX39B mRNAs but not the

trans-gene DDX39B mRNAs. Depletion or rescue of DDX39B protein was confirmed by western blot using anti-UAP56 (DDX39B) antibody (ab18106, Abcam) and α-Tubulin (AB_1904178, Cell Signaling Technology) as loading control.

## RNA-seq library preparation, sequencing and analyses

Total RNA was isolated from control (NTC) or DDX39B-depleted (Sh3 or Sh5) CD4+ T cells using ReliaPrep RNA Cell Miniprep System (Promega), and treated in-column with DNase I following the manufacturer's recommendations. Poly-A+ RNA was enriched from 1 µg of total RNA and used as template to generate libraries using the Illumina TruSeq platform as recommended by the manufacturer. Libraries were sequenced on a 2x100 paired-end format on an Illumina Hi-Seq 1500. Reads were aligned to the human GRCh38 reference with program STAR version 2.5.2b, using the parameters recommended for the ENCODE consortium (*ENCODE Project Consortium, 2011*; *Dobin et al., 2013*). The STAR genome index was built from the GENCODE primary genome assembly and the corresponding primary annotation file. Gfold software, version 1.1.4, with default parameters was used to count reads per gene and estimate expression differences between treatments. The GENCODE V24 basic annotation file was used for the gene counts. In order to determine significantly changed transcript abundance, an initial cutoff was used to discard transcripts averaging an RPKM of less than 2, then with a secondary cutoff of |GFold value|>0.3 or>0.1 (for Sh3 or Sh5, respectively, compared to NTC from the same donor). A lower cutoff was used for Sh5 since it was less effective than Sh3 at knocking down DDX39B. Finally, only transcripts fulfilling these parameters in both Donor 1 and Donor 4 were included as significantly changed (see *Supplementary file 1*).

Splicing analysis was carried out using Vast-tools program version 0.2.1 (*Irimia et al., 2014*) by aligning the paired-end reads to the Vast-tools human database (vastdb.hsa.7.3.14) using the default parameters. The Sh3-treated sample from each donor were compared using the differential function of vast-tools to determine changes in splicing relative to NTC in the same donor. Subsequently, we used the minimum value of change percent spliced in at 95% confidence interval value (MV|dPSI|; https://github.com/vastgroup/vast-tools, copy archived at swh:1:rev:fb87ceb4379cd9fbd-36fc402e4a3429567021548; *Irimia, 2023*) to filter for significantly changed events using 0.15 as cutoff (*Supplementary file 6*).

## MS susceptibility gene enrichment analysis

The study by the International Multiple Sclerosis Genetics (IMSG) Consortium (*Patsopoulos et al., 2019*) identified single nucleotide polymorphisms (SNPs) genetically and/or functionally associated with increased MS risk, from which a map of MS susceptibility genes was determined. We created a functional classification of the MS susceptibility genes (the genes with exonic or intronic SNPs, or cis-eQTL, MS_Susceptibility: 558 genes; see supplemental tables 7 & 19 in *Patsopoulos et al., 2019*), in lymphocytes based on expression quantitative trait loci (eQTL). Gene expression profiles of the MS susceptibility genes in EBV-transformed lymphocytes were obtained from GTExPortal (*GTEx Consortium, 2017*) and merged with data of odds ratios (OR) of MS risk to generate risk SNP and susceptibility gene pairs. Pairs of SNPs and genes with missing expression profiles were excluded from the analysis. A list of 539 pairs of SNPs and genes was generated (*Supplementary file 3*). If the MS risk allele (OR >1) showed higher expression of the corresponding susceptibility gene (Normalized expression: NES <0) in the lymphoblastoid cells, that gene was defined as a pathogenic MS gene (MS_Pathogenic: 250 genes). If the MS risk allele (OR >1) showed lower expression of the corresponding susceptibility gene (NES >0), that gene was defined as a protective MS gene (MS_Protective: 262 genes).

In order to estimate enrichment of differentially expressed genes (DEGs) in pre-determined MS gene sets, we used a simulation method. We determined DEGs in two donors with the shRNA that effectively knocked down DDX39B, Sh3, based on the Gfold method (*Feng et al., 2012*) with a cutoff of Gfold value change >0.3. The observed number of DEGs found in the MS gene set, k, was calculated by intersecting DEGs with each MS gene set (MS_Susceptibility, MS_Pathogenic, or MS_Protective). We then calculated enrichment using a resampling method. We constructed a sampling distribution for the null hypothesis (X) by randomly resampling $n_1$ genes from N, and counting the number of genes overlapping with each MS gene list $n_2$ over 100,000 permutations. An empirical p

value was calculated based on the fraction of instances that simulated (X) was greater than or equal to the actual observed value of k for each MS gene set.

## RT-qPCR analysis of RNA expression

Total RNA was isolated from control or DDX39B-depleted cells using ReliaPrep RNA Cell Mini-prep System (Promega) or Direct-zol RNA kit (Zymo Research), and treated in-column with DNase I following the manufacturer's recommendations. Reverse transcription was conducted with random primers using the High Capacity cDNA Reverse Transcription Kit (Thermo Fisher Scientific). DDX39B and FOXP3 RNA levels were measured by real-time quantitative PCR (RT-qPCR) using PowerUP SYBR Green Master Mix (Thermo Fisher Scientific) in a StepOnePlus Real-Time PCR System (Thermo Fisher Scientific). *EEF1A1* was used as normalization control since its expression was not affected by DDX39B depletion.

Genes whose expression was found to be differentially expressed upon DDX39B depletion in the RNAseq dataset in primary CD4[+] T cells were analyzed by RT-qPCR in control (NTC) or DDX39B-depleted (Sh3 and Sh5) primary CD4[+] T cells (from 6 donors), MT-2 cells and primary Tregs (from 2 donors). Target genes and primers are shown in *Supplementary file 8*. The data were normalized to *EEF1A1* expression. Differentially expressed genes were determined by comparing levels in DDX39B-depleted cells versus control cells.

## Western blot analysis of DDX39B and FOXP3 protein expression

Whole cell lysates from control (NTC) or DDX39B-depleted (Sh3 and Sh5) cells were collected in 1 X RIPA buffer (150 mM NaCl, 1% NP-40, 0.5% sodium deoxycholate, 0.1% SDS, and 50 mM Tris-HCl at pH 7.5) freshly supplemented with 1 X protease inhibitors (Roche). The cell lysates were quantified by bradford and equal amount of total protein was loaded per lane on NuPAGE 4–12% Bis-Tris pre-cast gels (Life Technologies), transferred to nitrocellulose membranes (Whatman), and blotted using standard protocols with antibodies against DDX39B (ab18106, Abcam), FOXP3 (AB_467554, Thermo Fisher Scientific), and either PTB, α-Tubulin (AB_1904178, Cell Signaling Technology) or α-beta-actin (sc-47778, Santa Cruz) as loading control.

## Gene-set enrichment analysis (GSEA)

To conduct GSEA analysis with our RNAseq data set in primary CD4[+] T cells, we generated a gene list from the control condition (NTC: Donor1-NTC, Donor4-NTC) and a gene list from the DDX39B knock-down condition (KD: Donor1-Sh3, Donor4-Sh3). All the genes that passed the initial cutoff (RPKM of ≥2 in both NTC libraries) were used. GSEA was conducted with the C2 curated genesets collections of Molecular Signatures Database (*Subramanian et al., 2005*). Among 5501 genesets in the C2 collection, 730 genesets passed the cutoff of the nominal p-value (*P*<0.05). Of these, 648 and 82 genesets showed enrichment to NTC and KD groups, respectively (*Supplementary file 4*).

## Subcellular fractionation of MT-2 cells

Subcellular fractionation of MT-2 cells was conducted following a protocol described previously (*Bhatt et al., 2012*) with minor modification. Control (NTC) or DDX39B-depleted (Sh3 and Sh5) MT-2 cells were washed with Phosphate-buffered saline (PBS), then lysed in NP40 cytoplasmic lysis buffer [0.075% (v/v) NP-40, 20 mM Tris-HCl pH7.5, 150 mM NaCl, 1 mM DTT, and 1 X protease inhibitor] for 2.5 minutes on ice. Cell lysates were then layered on top of a sucrose cushion [24% (w/w) sucrose, 20 mM Tris-HCl pH7.5, 150 mM NaCl, 1 mM DTT, and 1 X protease inhibitor] and centrifuged at 14,000 rpm for 10 min. For further separation of nucleoplasm and chromatin, the pelleted nuclei were treated with glycerol nucleoplasm lysis buffer [50% (v/v) Glycerol, 20 mM Tris-HCl pH7.5, 75 mM NaCl, 0.5 mM EDTA, 1 mM DTT and 1 X protease inhibitor], mixed gently, and treated with urea nuclei lysis buffer [1% (v/v) NP-40, 1 M Urea, 20 mM HEPES pH7.5, 1 mM DTT, 7.5 mM MgCl2, 0.2 mM EDTA and 1 X protease inhibitor] for 2 min on ice, followed by centrifugation at 14,000 rpm for 2 min. Supernatants were collected as nucleoplasm fraction and the pellets were collected as the chromatin fraction. Successful separation of subcellular compartments was confirmed by western blot with antibodies against markers of the different compartments: cytoplasmic α-Tubulin (AB_1904178, Cell Signaling Technology), nucleoplasmic Nucleolin (AB_533406, Bethyl Laboratories), and chromatin-associated hnRNP C (AB_627731, Santa Cruz Biotechnology). RNA was isolated from each fraction

using Direct-zol RNA kit (Zymo Research), and 200 ng of RNA from each compartment was used for reverse transcription with random primers as above. The abundance of FOXP3 and EEF1A1 RNAs were quantified in each fraction by RT-qPCR and normalized to total RNA as above. The percentage of EEF1A1 and FOXP3 RNA in each compartment was calculated by dividing by the corresponding total signal from the three compartments.

## RT-qPCR analysis of retained introns in *FOXP3*, *FOXP1* and other RNAs

To quantify intron retention events, we designed RT-qPCR primers to amplify total FOXP3 transcripts (constitutive exon 11), spliced FOXP3 transcripts (spanning the exon-exon junction between constitutive exons 10 and 11) and retained intron-containing FOXP3 transcripts: intron 2 (spanning the intron 2/exon 3 junction), intron 4 (spanning the intron 4/exon 5 junction), intron 6 (spanning the exon 6/intron 6 junction), intron 7 (spanning the intron 7/exon 8 junction), intron 9 (spanning the exon 9/intron 9 junction), or intron 11 (spanning the intron 11/exon 12 junction) (*Supplementary file 8*). Likewise, we designed primers to amplify total FOXP1 transcripts (within constitutive exon 21) and FOXP1 transcripts with retained introns: intron 11 (spanning intron 11/exon 12 junction), intron 14 (spanning intron 14/exon 15 junction), or intron 19 (spanning intron 19/exon 20 junction). Retained intron values were normalized to the corresponding total FOXP3 or FOXP1 RNA, and the data for DDX39B-depleted (Sh3 or Sh5) primary CD4+ T cells, MT-2 cells or primary Tregs is presented as fold-change over control (NTC). For the rescue experiments of endogenous DDX39B-depletion in MT-2 cells, cells transduced with NTC and GFP lentiviruses were used as controls. Detections of retained introns in FAM3A, PORCN, RBM10, RENBP, CFP and G6PD (C-rich py tracts) or FMR1 and DDX3X (U-rich py tracts) were conducted in the same manner. All primers used for this analysis are shown in *Supplementary file 8*.

## Phylogenetic analysis of the *FOXP3* gene

The evolutionary history of the *FOXP3* gene was inferred by using the Maximum Likelihood method based on the Tamura-Nei model (*Tamura and Nei, 1993*), and a phylogenic tree was generated with MEGA7 software (*Kumar et al., 2016*). The percentage of trees in which the associated taxa clustered together is shown next to the branches. Initial tree(s) for the heuristic search were obtained automatically by applying Neighbor-Join and BioNJ algorithms to a matrix of pairwise distances estimated using the Maximum Composite Likelihood approach, and then selecting the topology with superior log likelihood value. The tree is drawn to scale, with branch lengths measured in the number of substitutions per site. The analysis involved 15 genomic sequences of *FOXP3*, *FOXP3a* and *FOXP3b* gene from different vertebrate organisms: Human_ENSG00000049768, Gorilla_ENSGGOG00000000913, Macaque_ENSMMUG00000008624, Tarsier_ENSTSYG00000035932, Microbat_ENSMLUT00000005714.2, Mouse_ENSMUSG00000039521, Rat_ENSRNOG00000011702, Cow_ENSBTAG00000013279, Dog_ENSCAFG00000015934, Elephant_ENSLAFG00000003504, Opossum_ENSMODG00000009847, Platypus_ENSOANG00000013584, *Xenopus*_ENSXETG00000031498, Zebrafish_FOXP3a_ENSDARG00000055750, and Zebrafish_FOXP3b_ENSDARG00000078279.

Likewise, the evolutionary history of the *FOX* family gene was inferred by using the Maximum Likelihood method as described above. The analysis involved 12 protein-coding sequences of human FOX family genes (FOXA1: ENST00000250448.3, FOXD1: ENST00000615637.3, FOXE1: ENST00000375123.4, FOXF1: ENST00000262426.6, FOXH1: ENST00000377317.4, FOXN1: ENST00000226247.2, FOXO1: ENST00000379561.6, FOXP1: ENST00000318789.9, FOXP2: ENST00000350908.9, FOXP3: ENSG0000004976, FOXP4: ENST00000373063.7, and RBFOX2: ENST00000449924.6).

## FOXP3 splicing reporter assays

A total of $2.0 \times 10^5$ HeLa cells were transfected in 12-well plates with control (NTC: all-stars non-targeting control, Qiagen) or DDX39B-targeting (D11: Hs_BAT1_11, and D13: Hs_BAT1_13, Qiagen) siRNAs using Lipofectamine RNAiMAX Transfection Reagent (Thermo Fisher Scientific). After two days of culture, the Renilla luciferase (RLuc) splicing reporter plasmids were co-transfected with a Firefly luciferase plasmid (pGL3 control-FLuc, Promega) as transfection control. After 24 hr, cell lysates were collected for measurements of luciferase activity and RNA isolation. RLuc and FLuc activities

were measured by Dual-Luciferase Reporter Assay System (Promega), and the data are presented as RLuc/FLuc.

To directly quantify the splicing efficiency of the reporters, RNA was isolated and used for reverse transcription as described above. Spliced and unspliced reporter transcripts were measured by endpoint PCR (Platinum Taq DNA polymerase, Thermo Fisher Scientific) with primers specific to the RLuc coding sequence. PCR amplicons were detected by electrophoresis on 6% non-denaturing poly-acrylamide/TBE gels with SYBR Gold Nucleic Acid Gel Stain (Thermo Fisher Scientific). Splicing efficiency was measured by densitometry analysis of the PCR amplicons in image J (*Schindelin et al., 2012*; *Schneider et al., 2012*).

For the splicing-rescue assay, HeLa-Flp-In cells stably expressing either wild-type or mutant (K95A, E197A or D199A) DDX39B *trans*-genes were transfected with control siRNA (NTC) or DDX39B-targeting siRNA D13, which targets the 3' untranslated region (3'UTR) of DDX39B mRNA. After two days of culture, expression of DDX39B *trans*-genes in NTC- or D13-treated cells was either induced by 1 µg/mL doxycycline or not (doxycycline withheld) to generate the following four conditions: NTC −doxycycline (control), NTC +doxycycline (overexpression), D13 −doxycycline (knockdown) and D13 +doxycycline (rescue). The cells were then transfected with pcDNA3.1-RLuc_FOXP3 Intron 11 and harvested 24 hr after for analyses. DDX39B protein expression was confirmed by western blot, and the splicing efficiency of the reporters was evaluated by endpoint PCR as described above.

## Py tract sequence analysis of retained introns

5' SS (from −3 to +6 nt at the 5' Exon-Intron junction) and 3' SS (from −20 to +3 nt at the 3' Intron-Exon junction) sequences of *FOXP3*, *FOX* family genes and X chromosome genes (2155 genes) were obtained from Human genome assembly (GRCh38.p13) using the UCSC table browser (*Karolchik et al., 2004*; *Kent et al., 2002*) and Galaxy platform (*Afgan et al., 2018*). Sequence logos of the splice sites of *FOX* family genes and X chromosomal genes (total 2155 genes) were generated by WebLogo3 (*Crooks et al., 2004*). Sequences of 3' SS (from −20 to −3 nt) were defined as poly-pyrimidine tract (Py tract) sequence, and nucleotide probability at each of these positions was determined per gene. The sequence logo for a given gene shows the average at each position of the Py tract from all introns for that particular gene.

For comparison of Py tracts of introns that were DDX39B-sensitive (retained in the knockdown) versus DDX39B-insensitive (unchanged in the knockdown), we compared the 402 introns that were significantly retained versus 500 events randomly selected from the total unchanged introns. 5' SS and 3' SS sequences for these events were obtained using the Galaxy platform, and MaxEntScore for these were determined by MaxEntScan (*Yeo and Burge, 2004*). The nucleotide probability at each position of the Py tract region was determined as above.

## Statistical analysis

In all figures, error bars represent standard deviation (S.D.) unless otherwise noted. Asterisks denote level of statistical significance: * $p<0.05$, ** $p<0.01$, *** $p<0.001$ and **** $p<0.0001$. Statistical analyses were conducted in Prism8 (GraphPad Software) or SAS (SAS Institute). For the statistical analysis of experiments using CD4[+] T cells from individual donors, One-way repeated measures ANOVA was used. To specifically compare each experimental (Sh3 and Sh5) group to control (NTC) after a significant Omnibus f test, Dunnett's test was performed to control family-wise type I error rate due to multiple comparisons. For the experiments other than CD4[+] T cells, One-way ANOVA model followed by aforementioned Dunnett's tests were conducted to compare between experimental (Sh3 and Sh5) and control (NTC). For experiments done using iTregs from Donor 10, no statistical methods were used due to the difference in number of replicates for each experimental condition (NTC: n=2; Sh5: n=3). For the rescue assay of splicing, One-way ANOVA models followed by Tukey's tests were used to perform pairwise comparisons among all tested conditions. For the sequence analysis of retained introns, outliers of Max entropy scores were identified and removed (ROUT, Q=0.5%) (*Motulsky and Brown, 2006*). Then, Mann-Whitney tests or Kruskal-Wallis tests were conducted.

Statistical analysis of RT-qPCR data in CD4[+] T cell were carried out using a linear mixed model on the delta CT values (dCT). For each gene, the linear mixed model includes treatment (Control, DDX39B Knockdown1 [Sh3], DDX39B Knockdown2 [Sh5]) as fixed effect, and random intercept to account for the heterogeneity of the individuals and correlations among repeated measures from

the same individual. The fold-change differences (delta-delta CT [ddCT] values) and their standard error (SE) between treatment groups versus the control were estimated from the model. The p-values were adjusted by False Discovery Rate (FDR) method, which controls the false discovery rate due to multiplicity in hypothesis testing. To facilitate the interpretation using relative expression, the point estimate of the ddCT values were converted to $2^{-ddCT}$.

## Acknowledgements

We thank Drs. Paul Boutz (Rochester) and Harinder Singh (Pittsburgh) for excellent suggestions on an early draft of this manuscript. We acknowledge members of our laboratories for excellent discussions and comments to improve the science and the manuscript (MGB, SSB, SGG, DCK). WSF thanks John Paul Donohue (RNA Center for Molecular Biology, UC-Santa Cruz) for expert advice on the analysis of RNA sequencing data. Funding: We acknowledge support from the Uehara Foundation Fellowship and McLaughlin Postdoctoral Fund (MH), NIH F32 NS087899 (GGM), Duke Neurology startup and Stone family funds (SGG), KL2 TR001441-07 (WSF), R21AI133305 (DCK and LW), Duke MGM Start-up funds (DCK), R01 CA204806 (MGB), UTMB startup funds (MGB), and P01 AI150585 (MGB).

## Additional information

### Competing interests

Gaddiel Galarza-Muñoz, Mariano A Garcia-Blanco: I acknowledge that I have significant ownership in Autoimmunity Biologic Solutions, Inc (Galveston, TX), which is commercializing therapies that target the IL7R pathway in autoimmune diseases. While I do not believe this represents a conflict of interest it can lead to the perception of said conflict. The other authors declare that no competing interests exist.

### Funding

| Funder | Grant reference number | Author |
| --- | --- | --- |
| Uehara Foundation | | Minato Hirano |
| McLaughlin Family Foundation | | Minato Hirano |
| National Institutes of Health | NIH F32 NS087899 | Gaddiel Galarza-Muñoz |
| Stone Family Fund | | Simon G Gregory |
| National Institutes of Health | KL2 TR001441-07 | William S Fagg |
| National Institutes of Health | R21AI133305 | Dennis C Ko |
| National Cancer Institute | R01 CA204806 | Mariano A Garcia-Blanco |
| National Institutes of Health | P01 AI150585 | Mariano A Garcia-Blanco |

The funders had no role in study design, data collection and interpretation, or the decision to submit the work for publication.

### Author contributions

Minato Hirano, Conceptualization, Data curation, Formal analysis, Investigation, Writing – original draft, Writing – review and editing; Gaddiel Galarza-Muñoz, Conceptualization, Data curation, Formal analysis, Funding acquisition, Investigation, Writing – review and editing; Chloe Nagasawa, Formal analysis, Investigation, Writing – review and editing; Geraldine Schott, Investigation, Writing – review and editing; Liuyang Wang, Steven G Widen, Data curation, Formal analysis, Investigation; Alejandro L Antonia, Vaibhav Jain, Investigation; Xiaoying Yu, Data curation, Formal analysis, Methodology, Writing – review and editing; Farren BS Briggs, Formal analysis, Writing – review and editing; Simon G

Gregory, Conceptualization, Supervision, Writing – review and editing; Dennis C Ko, Conceptualization, Formal analysis, Supervision, Writing – review and editing; William S Fagg, Data curation, Formal analysis, Supervision, Investigation, Writing – review and editing; Shelton Bradrick, Conceptualization, Formal analysis, Supervision, Investigation, Writing – review and editing; Mariano A Garcia-Blanco, Conceptualization, Formal analysis, Supervision, Funding acquisition, Writing – original draft, Project administration, Writing – review and editing

Author ORCIDs
Steven G Widen http://orcid.org/0000-0002-6512-363X
Dennis C Ko http://orcid.org/0000-0002-0113-5981
Shelton Bradrick http://orcid.org/0000-0003-1566-9797
Mariano A Garcia-Blanco http://orcid.org/0000-0001-8538-1997

Decision letter and Author response
Decision letter https://doi.org/10.7554/eLife.76927.sa1
Author response https://doi.org/10.7554/eLife.76927.sa2

## Additional files

### Supplementary files

• Supplementary file 1. Differentially expressed genes after DDX39B-depletion observed with both shRNAs in both donors. Related to *Figure 1* and *Figure 1—figure supplement 1*.

• Supplementary file 2. Differentially expressed genes after DDX39B-depletion (Sh3) observed in both donors. Related to *Figure 1* and *Figure 1—figure supplement 1*.

• Supplementary file 3. List of MS Susceptibility SNPs and genes analyzed. Related to *Figure 1* and *Figure 1—figure supplements 2 and 3*.

• Supplementary file 4. Gene sets enriched in cells with normal levels of DDX39B (NTC). Related to *Figure 3* and *Figure 3—figure supplement 1*.

• Supplementary file 5. Connectivity Map analysis of the DDX39B KD. Related to *Figure 3* and *Figure 3—figure supplement 1*.

• Supplementary file 6. VastTool analysis of in CD4+T cells with DDX39B-depletion. Related to *Figure 5* and *Figure 5—figure supplement 3*.

• Supplementary file 7. Human and Mouse FOXP3 introns. Related to *Figure 6*.

• Supplementary file 8. Primer and Gblock sequences used in this study.

• Transparent reporting form

### Data availability

There will be no restriction on any material described in this manuscript. The bulk RNA sequencing datasets have been uploaded to the Gene Expression Omnibus with accession number GSE145773. All data are available in the main text or supplementary materials.

The following dataset was generated:

| Author(s) | Year | Dataset title | Dataset URL | Database and Identifier |
|---|---|---|---|---|
| Garcia-Blanco MA | 2023 | The RNA helicase DDX39B activates FOXP3 RNA splicing to control T regulatory cell fate | http://www.ncbi.nlm.nih.gov/geo/query/acc.cgi?acc=GGSE145773 | NCBI Gene Expression Omnibus, GSE145773 |

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
