## [Editor Report]

DDX39B is a helicase with known functions in mRNA splicing and nuclear export. This important study provides convincing evidence that DDBX39B regulates Foxp3, a lineage marker for T-regulatory cells in the immune system. The work provides a detailed analysis of the post-transcriptional regulation of Foxp3, and also positions DDX39B more broadly, as an important player in the regulation of autoimmune responses. The work will be of interest to RNA biologists, immunologists, and those studying autoimmune disorders.

---

## [Decision Letter]

**Decision letter after peer review:**

Thank you for submitting your article "The RNA helicase DDX39B activates FOXP3 RNA splicing to control T regulatory cell fate" for consideration by *eLife*. Your article has been reviewed by 2 peer reviewers, and the evaluation has been overseen by a Reviewing Editor and Satyajit Rath as the Senior Editor. The reviewers have opted to remain anonymous.

Essential revisions:

Specific concerns and suggested experiments are as below.

1. Figure 1E: Data from more than one donor should be shown and individual donors linked as Figure 1C. The same applies to Figure 2 C-F.

The inclusion of non-transduced Tregs, as an internal control from the same culture conditions is an important control. One way to do this is as follows: If pCLE-GFP was the marker for transduced cells, Foxp3 MFI should be reported in GFP- and GFP+ gates, to demonstrate that Foxp3 expression is indeed reduced upon knock-down of DDX39B.

2. Figure 2: Is the GSEA analysis performed from only DDX39B knocked-down Treg dataset? Or from all the T helper types?

Foxp3 dependent gene expression should be maximally affected in the Tregs upon DDX39B knockdown if it is an effect mediated through Foxp3. If the same extent of reduction is observed in both CD4 cells (Tregs and non-Tregs), then that would suggest that many of these genes are themselves under DDX39B control, independent of Foxp3. In fact, this second possibility seems to be likely, especially for IL10, due to two reasons. First, the extent of its downregulation upon DDX39B knock-down appears to be the same in pan CD4 T cells and primary Tregs, if anything, it is much lower in pan T-cells. Second, IL10 is not a direct target of Foxp3, since in Foxp3 deficient Foxp3GFPKO mice, IL10 expression is not affected (Gavin 2007 Nature; doi: 10.1038/nature05543). So the conclusion that IL10 downregulation is indicative of reduced Foxp3 function merits re-examination. The possibility that IL10 itself is a target of DDX39B, which may contribute to MS susceptibility may also be considered. Treg functionality upon DDX39B knockdown can be assessed by in vitro suppression assay. If indeed Foxp3 downregulation is the consequence, this is expected to directly affect Treg mediated suppressive capacity.

3. Figure 3: Key conclusions be validated in in vitro TGF β generated iTreg cells, where enough cells are expected to be generated for FACS analyses and real-time PCR mediated intron retention assays.

4. Figure 6: Β-globin intron, as mentioned by authors, is unaffected. Does splitting and the luciferase signal remain unchanged upon DDX39B knockdown. These experiments would be most convincing if done using iTregs

5. The lack of mechanistic understanding of why certain C-rich introns are sensitive to DDX39B is a key gap. However, the authors can address whether DDX39B binds better to introns it regulates vs. others – either in vitro (EMSA) or by CLIP-Seq.

Comments not requiring new experiments:

1. Figure 3: At first, the dramatic reduction of Fox3p in the cytoplasm in Sh3/5-treated cells seems to suggest a role of DDX39B in nuclear export of Fox3p (in addition to, or instead of early processing). Upon further reflection, the argument against this conclusion is that the nucleoplasmic pool is not increased significantly – rather only the chromatin-associated pool. The authors should clarify this point.

In the description in line numbers 208-213, Figure S3C, which cells are used for this experiment?

2. Figure 4: The rescue experiments are commendable and convincing. However, the transgene clearly doesn't recover full splicing or expression of Fox3p even though higher than endogenous levels of DDX39B are achieved. Some comment on what this might be due to or imply would be useful.

The authors may consider making a subfigure after customizing the height of the tracks so that the difference between control and DDX39B knockdown becomes more apparent.

3. Figure 5A: The rationale for comparing Fox3p exons to those of other genes specifically on the X chromosome is unclear. Would a different result be obtained if compared to all genes?

---

## [Author Response]

Essential revisions:Specific concerns and suggested experiments are as below.1. Figure 1E: Data from more than one donor should be shown and individual donors linked as Figure 1C. The same applies to Figure 2 C-F.

In response to concern about original Figure 1E and based on comments below re the advantage of using iTreg cells, we repeated critical experiments in iTregs from two additional human donors. Data from Tregs from four human donors are now shown in new Figure 2.

The inclusion of non-transduced Tregs, as an internal control from the same culture conditions is an important control. One way to do this is as follows: If pCLE-GFP was the marker for transduced cells, Foxp3 MFI should be reported in GFP- and GFP+ gates, to demonstrate that Foxp3 expression is indeed reduced upon knock-down of DDX39B.

We respectfully disagree with the reviewer on the use of non-transduced Tregs since it is known that transduction can lead to spurious effects that must be controlled. For transduction experiments, including those shown in original Figure 2D, we used lentivirus coding non-targeting shRNA as control, thus, we cannot use the GFP signal as a transduction control. Also, we selected the cells with puromycin after transduction, so it is unlikely to be biased by the non-transduced cells. Orginal Figure 2D showed IL2RA, not FOXP3.

2. Figure 2: Is the GSEA analysis performed from only DDX39B knocked-down Treg dataset? Or from all the T helper types?

The GSEA analysis, which is now shown in new Figure 3, was derived from data from DDX39B kd vs control in CD4^+^ Tcells. To make sure this is clear we carefully edited the Results section and the legend to new Figure 3.

Foxp3 dependent gene expression should be maximally affected in the Tregs upon DDX39B knockdown if it is an effect mediated through Foxp3. If the same extent of reduction is observed in both CD4 cells (Tregs and non-Tregs), then that would suggest that many of these genes are themselves under DDX39B control, independent of Foxp3. In fact, this second possibility seems to be likely, especially for IL10, due to two reasons. First, the extent of its downregulation upon DDX39B knock-down appears to be the same in pan CD4 T cells and primary Tregs, if anything, it is much lower in pan T-cells. Second, IL10 is not a direct target of Foxp3, since in Foxp3 deficient Foxp3GFPKO mice, IL10 expression is not affected (Gavin 2007 Nature; doi: 10.1038/nature05543). So the conclusion that IL10 downregulation is indicative of reduced Foxp3 function merits re-examination. The possibility that IL10 itself is a target of DDX39B, which may contribute to MS susceptibility may also be considered. Treg functionality upon DDX39B knockdown can be assessed by in vitro suppression assay. If indeed Foxp3 downregulation is the consequence, this is expected to directly affect Treg mediated suppressive capacity.

We thank the reviewer for this comment, with which we agree. We have revised the Results section and new Figure 3 carefully to make this point clear.

3. Figure 3: Key conclusions be validated in in vitro TGF β generated iTreg cells, where enough cells are expected to be generated for FACS analyses and real-time PCR mediated intron retention assays.

We thank the reviewer for this suggestion on data from original Figure 3 (new Figure 5) and indeed used iTregs from two human donors to repeat critical experiments done with Treg cells isolated from the first two human donors. *FOXP3* intron retention is exacerbated by DDX39B knockdown in Tregs/iTregs from all four human donors as shown in the new Figure 5—figure supplement 2. The result that several FOXP3 introns are highly susceptible to DDX39B levels was observed in CD4^+^ Tcells, the Treg-like MT-2 cells and in primary Tregs/iTregs, therefore we believe this critical conclusion is very strongly supported.

4. Figure 6: Β-globin intron, as mentioned by authors, is unaffected. Does splitting and the luciferase signal remain unchanged upon DDX39B knockdown. These experiments would be most convincing if done using iTregs

(Note: should refer to original Figure S4A) Since increased intron retention effect was observed in endogenous *FOXP3* introns in primary Tregs and iTregs form four human donors (new Figure 5—figure supplement 2), we did not repeat the reporter assays in iTregs.

5. The lack of mechanistic understanding of why certain C-rich introns are sensitive to DDX39B is a key gap. However, the authors can address whether DDX39B binds better to introns it regulates vs. others – either in vitro (EMSA) or by CLIP-Seq.

We agree with the reviewer about the importance of a mechanistic understanding of why certain introns with C-rich polypyrimidine tracts are more sensitive to DDX39B knockdown and indeed there are multiple projects in the laboratory investigating this issue. We respectfully submit that further investigation of this issue goes beyond the scope of the current manuscript. We point out that introns that bind DDX39B poorly will actually be the most sensitive to DDX39B depletion. It is these low affinity binding sites that are disproportionally affected when the nuclear DDX39B concentration is reduced by knockdown.

Comments not requiring new experiments:1. Figure 3: At first, the dramatic reduction of Fox3p in the cytoplasm in Sh3/5-treated cells seems to suggest a role of DDX39B in nuclear export of Fox3p (in addition to, or instead of early processing). Upon further reflection, the argument against this conclusion is that the nucleoplasmic pool is not increased significantly – rather only the chromatin-associated pool. The authors should clarify this point.

We have edited the text in a way that now should be much clearer (data now shown new Figure 4).

In the description in line numbers 208-213, Figure S3C, which cells are used for this experiment?

Figure S3C showed the result of analysis of NGS of CD4^+^ T cells.

2. Figure 4: The rescue experiments are commendable and convincing. However, the transgene clearly doesn't recover full splicing or expression of Fox3p even though higher than endogenous levels of DDX39B are achieved. Some comment on what this might be due to or imply would be useful.

We thank the reviewer and note that these rescue experiments are indeed challenging (new Figure 5). In the revised manuscript we have included an explanation in the Results section – we believe that once FOXP3 levels are reduced this leads to the disruption of epigenetic maintenance that makes recovery of FOXP3 expression difficult to achieve even when FOXP3 intron retention is rescued by DDX39B overexpression.

The authors may consider making a subfigure after customizing the height of the tracks so that the difference between control and DDX39B knockdown becomes more apparent.

Assuming this refers to original Figure 4 (new Figure 5) we respectfully suggest that the difference is clear as shown.

3. Figure 5A: The rationale for comparing Fox3p exons to those of other genes specifically on the X chromosome is unclear. Would a different result be obtained if compared to all genes?

We apologize for not making the rationale clear. *FOXP3* is on the X chromosome and we compared its introns to those of other protein-coding genes on the same chromosome. In the revised manuscript we re-wrote this to make the rationale clear. Based on work of many others that examined sequence composition of py tracts genomewide (e.g., Burge laboratory) leads to the same result as we observe in X chromosome genes. This is noted in the Results section associated with the data described in original Figure 5 (new Figure 6).